# Postural control of arm and fingers through integration of movement commands

**Scott T Albert[1]\*, Alkis M Hadjiosif[2], Jihoon Jang[1], Andrew J Zimnik[3], Demetris S Soteropoulos[4], Stuart N Baker[4], Mark M Churchland[3], John W Krakauer[2], Reza Shadmehr[1]\***

[1]Department of Biomedical Engineering, Johns Hopkins School of Medicine, Baltimore, United States; [2]Department of Neurology, Johns Hopkins School of Medicine, Baltimore, United States; [3]Department of Neuroscience, Columbia University, New York, United States; [4]Institute of Neuroscience, Newcastle University, Newcastle upon Tyne, United Kingdom

**Abstract** Every movement ends in a period of stillness. Current models assume that commands that hold the limb at a target location do not depend on the commands that moved the limb to that location. Here, we report a surprising relationship between movement and posture in primates: on a within-trial basis, the commands that hold the arm and finger at a target location depend on the mathematical integration of the commands that moved the limb to that location. Following damage to the corticospinal tract, both the move and hold period commands become more variable. However, the hold period commands retain their dependence on the integral of the move period commands. Thus, our data suggest that the postural controller possesses a feedforward module that uses move commands to calculate a component of hold commands. This computation may arise within an unknown subcortical system that integrates cortical commands to stabilize limb posture.

**\*For correspondence:**
scottalbert1@gmail.com (STA);
shadmehr@jhu.edu (RS)

**Competing interests:** The authors declare that no competing interests exist.

## Introduction

To hold the limb still, the muscles are not quiet. Rather, they are actively engaged with coordinated inputs that maintain postural stability. Current models assume that these inputs are produced by an impedance controller that translates the sensory representation of a desired location to patterns of muscle activity (*Yadav and Sainburg, 2011*; *Lametti et al., 2007*; *Schabowsky et al., 2007*). To move and then hold, one feedback controller generates the commands that move the limb (*Todorov and Jordan, 2002*; *Liu and Todorov, 2007*), and a separate controller generates commands that hold the limb still following movement (*Yadav and Sainburg, 2011*). This architecture (*Figure 1A*, left) in which movement and postural controllers are independent is implicit in optimal control formulations of reaching (*Yadav and Sainburg, 2011*; *Lametti et al., 2007*; *Todorov and Jordan, 2002*; *Ghez et al., 2007*) and forms the basis for interpreting how neurons in the motor cortex encode reach kinematics (*Sachs et al., 2016*). While many predictions of this theory have been confirmed for control of movement (*Todorov and Jordan, 2002*), here we provide evidence that challenges the assumption that posture and movement are controlled independently.

Our idea starts with consideration of a simpler control system: the eye and the head. In order to hold the eyes at a target, the oculomotor system uses a hold controller whose output directly depends on the move controller (*Cohen and Komatsuzaki, 1972*; *Cannon and Robinson, 1987*; *Crawford et al., 1991*; *McFarland and Fuchs, 1992*; *Miri et al., 2011*; *Shadmehr, 2017*; *Godaux et al., 1993*; *Cheron et al., 1986*). The move controller produces a set of commands that

**eLife digest** Moving an arm requires the brain to send electrical signals to the arm's muscles, causing them to contract. Neuroscientists call these types of brain signals "move signals". The brain also sends so-called hold signals, which hold the arm still in a desired position. Part of the brain known as the primary motor cortex helps to calculate the move signals for the arm, but it was unclear how the brain produces the corresponding hold signals.

Fortunately, the fact that the brain moves other things besides arms may help answer this question. Previous research has shown, for example, that a brain area called the "neural integrator" calculates the hold signals needed to hold the eye in a specific position. The neural integrator does this by using basic principles of physics, and details of the speed and duration of the eye's movements.

Now, Albert et al. show a similar mechanism appears to control hold signals for arm movements. In one set of experiments, muscle activity was measured as monkeys moved their arms or fingers to different target positions. In other experiments, human volunteers held a robot arm, and Albert et al. measured the forces they produced while reaching and holding still. Both the human and monkey experiments revealed a relationship between move signals and hold signals. Like for eye movements, hold signals for the arm could be calculated from the move signals. In further experiments with stroke patients where the brain had been damaged, the move signals were found to be deteriorated, but the way hold signals were calculated stayed the same. This suggests that there is an unknown structure within the brain that calculates hold signals based on move signals.

Investigating how the brain holds the arm still may help scientists understand why some neurological conditions like stroke or dystonia cause unwanted movements or unusual postures. This might also lead scientists to develop new ways to treat these conditions.

displace the eyes (*Shadmehr, 2017*). Simultaneously, these commands are integrated in real-time by a distinct brainstem structure, yielding sustained commands that hold the eyes and the head still when the movement ends (*Cohen and Komatsuzaki, 1972*; *Miri et al., 2011*). Thus, the architecture assumed for move and hold controllers of the arm (*Figure 1A*, left) is not consistent with that of the eye and the head (*Cohen and Komatsuzaki, 1972*; *McFarland and Fuchs, 1992*; *Godaux et al., 1993*; *Shadmehr, 2017*; *Cheron et al., 1986*; *Cannon and Robinson, 1987*).

While we do not know if the output of the reach controller serves as an input to the hold controller, there is evidence that moving and holding are controlled by separate neural structures; total inhibition of the mouse motor cortex during reaching causes the arm to stop moving, but the muscles continue to receive commands that hold the arm steady against gravity (*Guo et al., 2015*). It is difficult to reconcile this observation with the idea that the cortex drives both moving and holding (*Humphrey and Reed, 1983*; *Kurtzer et al., 2005*).

Yet, there are also reasons to doubt that the neural control of the arm shares a design principle with control of the eyes and the head. The physical dynamics of the arm are much more complicated than the eye, casting doubt that any straightforward relationship might exist between commands that move the arm to a location, and the commands that subsequently hold the arm there. Furthermore, whereas damage to the brainstem structure that holds the eyes produces nystagmus (*Kaneko, 1997*; *Kaneko, 1999*), we know of no condition that resembles nystagmus in the context of reaching.

We began by asking a simple question: are the commands that hold the limb at the target solely determined by the target position (*Figure 1A*, left), or are they dependent in part on the preceding move commands (*Figure 1A*, right)? We began by measuring activity across arm muscles during point-to-point reaching. When monkeys reached to a single target from various directions, we found that the integral of reach activity predicted hold activity after the movement ended. Furthermore, as the target location varied, the same integration function accounted for hold period activity at the various endpoints. Thus, across a range of directions and endpoints, the hold period activity was related, through integration, with the preceding move period activity.

To ask whether this pattern held across other types of movements, we considered goal-directed finger movements in which the start and end target locations were kept constant. As monkeys flexed

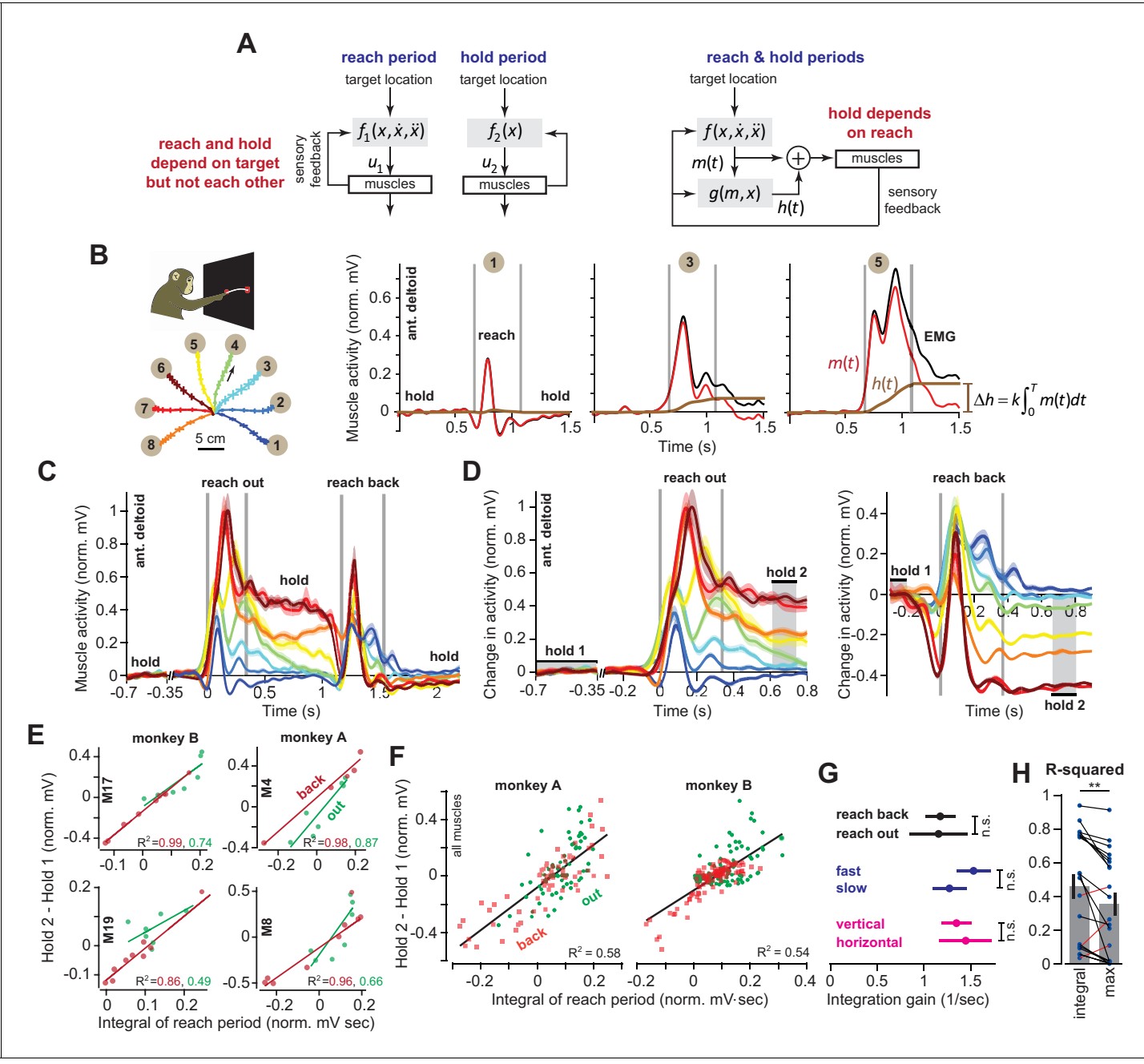

**Figure 1.** Integral of muscle activity during the reach correlates with subsequent activity during the hold period. (**A**) In current models (left), a feedback controller generates commands that move the arm, and then upon reach end, a postural controller holds the limb still. For this model, hold commands depend only on the target position. In the model considered here (right), the move commands are integrated in real-time by a postural controller. Thus, the hold commands depend on the preceding reach commands, not solely the target position. (**B**) Monkeys reached out to one of eight targets, waited, and then reached back to the home position. EMG from ant. deltoid is shown for three targets, and decomposed into $m(t)$ and $h(t)$ using *Equation (5)*, with $k = 1$. (**C**) Normalized activity of anterior deltoid in Monkey B starting from the center location. Colors correspond to targets in part B. (**D**) Change in ant. deltoid EMG from the initial hold period for reach out and reach back components of the task. The bars for hold 1 and hold 2 indicate periods where hold activity was calculated. (**E** and **F**) Change in hold period activity for reach out (green) and reach back (red) components of the task as a function of the integral of the preceding reach period in two representative muscles for each monkey, and all muscles. (**G**) The integration gain (slope of the line in E) across various conditions: outward vs. return, fast vs. slow, horizontal target positions vs. other targets. (**H**) Comparison of two hypotheses: hold period activity relates linearly to integral of previous reach period, or hold period activity relates linearly to the maximum activity of the muscle in the previous reach period. Each point is a single muscle. Error bars are SEM. Statistics: **$p<0.01$ and n.s. $p>0.05$.

The online version of this article includes the following source data and figure supplement(s) for figure 1:

*Figure 1 continued on next page*

*Figure 1 continued*

**Source data 1.** The EXCEL source data file contains holding EMG activity and the integral of moving EMG activity for all muscles and moving directions shown in *Figure 1E–H*.

**Figure supplement 1.** Muscles that do not integrate also do not contribute to posture.

their finger, there was natural variability in muscle activity both during the movement and during the ensuing hold period. However, this variability had structure: on a within-trial basis, changes in hold period commands in all recorded muscles were correlated with the integral of the preceding commands that had brought the finger to its current location.

These patterns revealed a correlation, not causation. To test whether there might be a causal link between movement and holding, we imposed a change to the commands that moved the arm to a given target location, and quantified whether on a within-trial basis, the change in move period commands influenced the subsequent hold period commands. To do this, we altered the reach commands of humans through adaptation (*Shadmehr and Mussa-Ivaldi, 1994*). To measure the properties of the hold controller, we designed a procedure in which we slowly displaced the hand during the hold period while subjects were engaged in a working memory task. We recorded the forces produced by the hand in response to the involuntary displacements during the hold period, thus measuring the postural field that held the arm still. We found that as the reach period commands changed, the entire postural field shifted, indicating that changes to reach commands altered the null point of the hold controller. Notably, the function that related the hold period controller to the preceding reach period was the same integration function that we had observed in point-to-point reaching and finger movements.

Finally, we probed the neural circuits that might support this putative link between movement and holding by examining reaching in patients who had suffered damage to their corticospinal tract (CST) above the level of the brainstem. As expected, these stroke survivors exhibited large trial-to-trial variability in the commands that they produced during both the reach and the subsequent hold periods. Remarkably, the link between move and hold period commands appeared intact: on a within-trial basis, the hold period forces were related via a form of integration to the immediately preceding, but now imperfect, reach period forces.

Thus, in monkeys, healthy humans, and stroke survivors, across arm movements and finger movements, the hold period commands depended on the preceding commands that had moved the limb to its current location. These results raise the possibility that the postural controller possesses a subcortical feedforward module that calculates hold period commands through real-time integration of the move period commands. This feedforward computation then combines with visual and proprioceptive feedback to produce the sustained commands that result in postural stability.

## Results

We performed experiments in which monkeys and humans made goal-directed movements toward a target and then held their arm or finger at the target location. In each case, we asked whether the commands during the move period influenced the subsequent commands during the hold period.

### Muscle activity during the hold period following reaching

When reaching movements are made from various starting points to the same target, we know of no model that predicts a consistent relationship between the reach period commands, which depend on reach direction, and the subsequent hold period commands, which depend on target location. Yet, if control of holding depends on the movement period (*Figure 1A*, right), a single function must exist that transforms the commands that were generated during the movement period to the subsequent commands that are generated to hold the arm. To explore the plausibility of such a relationship, we used intramuscular electrodes to measure activity of 20 shoulder and elbow muscles during point-to-point reaching in the vertical plane (nine muscles in monkey A, 11 muscles in monkey B). On each trial, monkeys first moved their hand from a central location to one of eight targets, held their hand at that target for at least 0.5 s, then reached back to the central location and again held their

hand for at least 0.5 s (*Figure 1B*). We normalized muscle activity by setting to zero its average activity at the central hold location, and setting to one its maximum activity in the task.

Consider the activity of the anterior deltoid (*Figure 1B*) as the arm reached from the central position to a target. For some directions, this muscle exhibited a burst of activity during the reach, and then sustained activity during holding (*Figure 1B*, targets 3 and 5). For other targets, the muscle exhibited a smaller burst of activity during movement, and little or no activity during holding (target 1). Let us imagine that the measured EMG, denoted as $u(t)$, is actually the sum of two signals: a 'hold' command $h(t)$ (brown traces in *Figure 1B*) and a 'move' command $m(t)$ (red traces in *Figure 1B*). The hold command is computed by adding the real-time integral of the move command to the initial hold activity that precedes movement (see Materials and methods Section A).

If the hold commands are computed from the move commands in this way, the two commands should be related by a common function across different types of reaching movements: the change in muscle activity from before the movement to after the movement, should be related to the intervening move period muscle activity (see Materials and methods A for derivation). We measured muscle activity with respect to its pre-movement hold period, $u(h_1)$, and then integrated that change with respect to time until the end of the reach, $t = T$.

$$u(h_2) - u(h_1) \approx k \int_{t_0}^{T} [u(t) - u(h_1)]dt + a \qquad (1)$$

This equation predicted that change in the hold period activity of a given muscle from before reach onset $h_1$ to the target $h_2$ should be approximately proportional to the integral of its activity during the reach period.

To test the validity of *Equation (1)*, we defined the hold period at the target $h_2$ starting at 300 ms after reach end (*Figure 1D*, hold 2), thus allowing time for muscle activation dynamics to settle. Indeed, for most muscles (14/20), change in the hold period activity was well predicted by *Equation (1)*. For example, when the target location was constant (*Figure 1D*, reach back) across various movement directions, the changes in the hold period activities of many muscles were proportional to the integral of their respective activity during the preceding reach (red lines, *Figure 1E*). When the target position varied (*Figure 1D*, reach out), *Equation (1)* was again a good predictor of the change in hold period activity, despite the fact that both direction and endpoint of the reach changed (green lines, *Figure 1E*).

*Figure 1F* presents the data across all muscles, directions, and endpoints. Remarkably, we found that integration of the reach period activity was a reasonable predictor of the change in hold period activity across all conditions ($R^2 = 0.58$ for Monkey A and $R^2 = 0.54$ for Monkey B). Within each muscle, the integration gain $k$ was no different for outward reaches and return reaches (*Figure 1G*, paired t-test on single muscle regression slopes, p=0.943). In addition, the same integration gain predicted hold activity for fast and slow movements (*Figure 1G*, fast vs. slow, two-sample t-test, p=0.30) which differed modestly in terms of movement duration (two-sample t-test, p<0.001, fast movement duration of 350.3 ± 11.1 ms and slow movement duration of 453.6 ± 4.9 ms, mean ± SEM). This indicated that a single function (*Equation (1)*) could account for various movements and speeds, despite the differing dynamics of these reaches.

Notably, despite these general trends, some muscles (6/20) did not exhibit the pattern described by *Equation (1)*. These muscles shared a specific property: they had little to no activity during the hold periods (*Figure 1—figure supplement 1*). Thus, *Equation (1)* seemed to apply primarily to those muscles that modulated their activity during the hold period, contributing to maintenance of arm posture in this task. However, to not bias our results, we included all muscles in our regressions in *Figures 1F, G and H*.

We considered an alternative hypothesis: a muscle that is more active to lift the arm will be also be more active to hold the arm. Perhaps, the correlations are driven mostly by biomechanical constraints of the reaching movements. Thus, we separated movements based on directions that were not affected by a change in gravitational forces (horizontal reaches), vs. other directions (*Figure 1B*). If the relationship between move period and hold period was solely due to the gravitational field, we would expect that the integration function would differ for horizontal versus vertical movements. However, the gain of integration was similar for the two groups of movements (*Figure 1G*, ANCOVA, movement type by moving EMG integral interaction effect on holding activity, F = 0.37,

p=0.54). To broaden the scope of this alternative hypothesis, we considered the possibility that the change in hold period activity in each muscle depended on the maximum or minimum activity of that muscle during the reach period, not the integral of the entire reach period activity. This alternative hypothesis also proved to be significantly less accurate than *Equation (1)* (within muscle comparison, paired t-test, p=0.005, *Figure 1H*). In 17/20 muscles, integration of the reach period activity was a better predictor of the hold period activity than either the maximum or minimum muscle activity.

In summary, for reaching across various directions and endpoints, the change in a muscle's activity from the pre-movement hold period to the post-movement hold period was partially predicted by the integral of that muscle's activity during the intervening reach period.

## Hold period activity for finger movements

According to *Equation (1)*, trial-to-trial variation in the move period commands should lead to consistent trial-by-trial changes in the subsequent hold period commands, even if the target location remains constant. That is, if the integral of a muscle's activity is greater on some trials, then that muscle should also be more active during the hold period that immediately follows.

It is difficult to test this prediction for reaching because there are numerous configurations of the wrist, elbow, and shoulder joints that would maintain the hand at a target location. Therefore, to more precisely examine within-trial covariance between move and hold periods, we simplified the problem to a single degree of freedom: finger flexion (*Soteropoulos et al., 2012*).

Monkeys were trained to use their index finger to track a visual target that moved at 12 deg/sec over a 1 s period between a start (12°) and an end location (24°) against a spring load that resisted flexion (*Figure 2A*). At the start location, the load was 0.026 N·m. As the finger flexed, the load increased linearly, reaching 0.048 N·m at the target. We measured muscle activity using subcutaneous electrodes implanted over 17 muscles (eight muscles in monkey R, nine in monkey D). For each session, we normalized each muscle's activity by setting the average activity at the start location to 0 (period $h_1$, 400 ms in duration, began 1 s before movement onset, *Figure 2C*), and setting its maximum activity in the task to 1. The hold period $h_2$ at the target was 200 ms in duration and began 700 ms after movement end, thus allowing time for muscle dynamics to settle.

The finger accurately tracked the target on each trial, moving along very similar trajectories (*Figure 2B*). To both flex the finger and support the mechanical load, muscle activity during each hold period was strongly modulated by finger location (*Figure 2C*). For example, the flexor digitorum profundus (FDP) muscle, one of the prime movers in this task, was more active when the finger was at the target as compared to the start location (*Figure 2C*, right panel). However, from one trial to the next, hold period EMG exhibited marked variability (*Figure 2C*, note the vertical spread of the trial distribution at hold 2).

One possibility is that the variability in hold period EMG is due to position-related variability in the spring force applied to the finger (larger displacements lead to larger spring forces on the finger). To assess this possibility, for each muscle we regressed its activity onto its position during the target hold period. We found that variation in hold position accounted for less than 2% ($R^2 = 0.0107 \pm 0.0041$, mean ± SEM across all muscles in both monkeys, individual regressions for each muscle) of the trial-to-trial variability in hold period EMG (we do not mean for this to give the impression that EMG was not strongly modulated by position, only that trial-by-trial differences in EMG were poorly explained by differences in position; see Materials and methods Section B2 for more information).

If variability in position could not explain the trial-to-trial changes in hold period EMG, what was the source of hold period variability? Inspection of muscle activity in *Figure 2C* suggested two possibilities: trial-by-trial changes in the hold period activity at the target (the hold 2 period) could be explained by (1) modulation in the initial holding activity prior to movement onset (the hold 1 period), or (2) modulation in move period activity. We first investigated the former possibility that when a muscle was more active during the initial hold period, it was also more active during the final hold period. This hypothesis stated that $u(h_2) \approx ku(h_1) + a$. To evaluate this hypothesis, we regressed the activity of each muscle during hold 2 onto its activity during hold 1. The left panel in *Figure 2D* shows the strength of this correlation in the FDP muscle for a single session. Each ellipse in the middle panel of *Figure 2D* represents the 95% confidence boundary for the trial-by-trial joint distribution between hold period activity at the target and hold period activity at the starting position, for

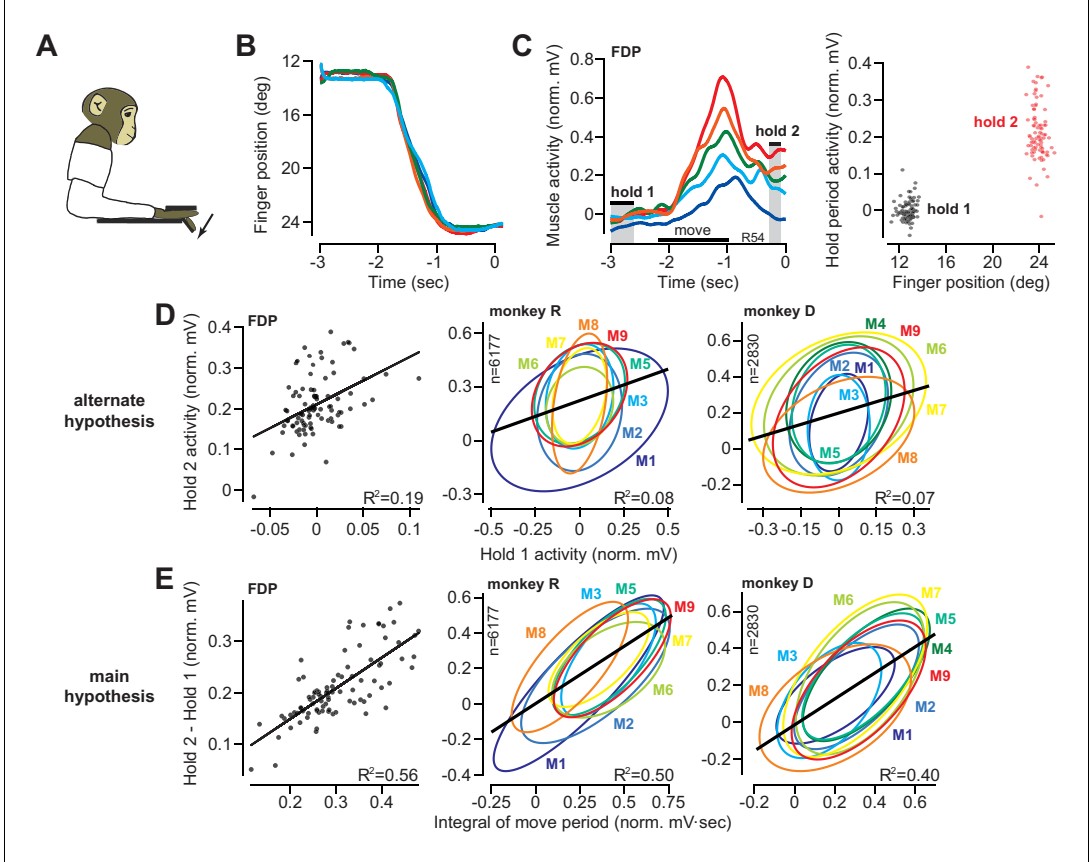

**Figure 2.** The integral of muscle activity during finger flexion correlates with subsequent activity during the hold period.  (A and B) Monkeys were trained to move their index finger from an initial position to a target against a load. The traces show representative movements. Positive displacements correspond to flexion. (C) Activity of flexor digitorum profundus (FDP) in monkey R. Left panel shows FDP activity for the trials shown in (B). The bars for hold 1 and hold 2 indicate periods was hold activity was calculated. Right panel shows FDP activity during the hold periods. Activity increased with flexion of the finger, but was variable from one trial to the next. (D) Evaluation of the hypothesis that variability in muscle activity at the hold 2 position could be explained by variability in the preceding hold 1 activity. Left panel is for the FDP muscle during a single session in monkey R. Center and right panels present data across all muscles recorded in each monkey. Each ellipse is the 95% confidence interval for a single muscle. $R^2$ value refers to a linear fit across all trials and muscles. (E) Same as for (D), except here we test the hypothesis that variability in hold period activity is related to the integral of the preceding moving activity.

The online version of this article includes the following source data for figure 2:

**Source data 1.** The EXCEL source data file contains holding EMG activity and the integral of moving EMG activity for all muscles shown in *Figure 2*.

each muscle, across all trials and sessions. Overall, hold activity at the start appeared to be a rather poor predictor of hold activity at the target, accounting for about 8% (monkey R) and 7% (monkey D) of the variance in the data (*Figure 2D*).

We next considered the hypothesis that variation in hold period activity may be due to variation in preceding move period activity. Using *Equation (1)*, we integrated the move period activity in the muscle, $u(t)$ with respect to its pre-movement activity, $u(h_1)$, and asked if this integral could predict the change in hold period activity from the start location, $u(h_1)$, to the target location, $u(h_2)$. This comparison could be confounded by trial-to-trial differences in movement displacement. Under spring forces, a larger displacement might lead to greater move period as well as greater hold period activity. This did not appear consistent with the data: trial-to-trial displacement explained less than 1% of the variance in both the integral of move period activity ($R^2 = 0.006 \pm 0.0018$, mean ± SEM), and the change in hold activity ($R^2 = 0.005 \pm 0.001$, mean ± SEM). On the other hand, the integral of move period activity exhibited robust correlation with the change in hold period activity, as shown for an example recording session in the left panel of *Figure 2E*. For this session, about

56% of the trial-by-trial variance in FDP hold period activity $u(h_2) - u(h_1)$ was accounted for by *Equation (2)* (linear fit for this session, p<0.001).

To determine if all muscles exhibited similar within-trial correlations between moving and holding, we considered the data across all muscles and sessions (*Figure 2E*, middle and right panels). Each ellipse in *Figure 2E* represents the 95% confidence boundary for the within-trial joint distribution between holding activity and the integral of moving activity, for each muscle. Strikingly, the orientations of various muscle distributions (the angle of the major axis) were roughly parallel with each other. Thus, the gain of the integration function was similar across muscles, and a single function accounted for approximately 50% and 40% (monkeys R and D) of the trial-by-trial variability in holding activity.

Finally, we considered another potential source of correlation between move and hold periods: co-contraction. If finger stiffness varied from one trial to another, we would observe correlations between move and hold periods. To change finger stiffness, agonist and antagonist pairs of muscles would exhibit coordinated increases or decreases in their activities. In other words, we should be able to predict the change in hold period activity in one muscle based on the activity of other muscles. To test this idea, we regressed the change in hold period activity in each muscle onto the integral of move period EMG in other muscles. Roughly 10% of the variability in the change in hold period EMG could be explained by the integral of move period activity of other muscles ($R^2 = 0.10 \pm 0.02$, mean ± SEM). Therefore, while some of the variance in hold period activity could be explained by the move period activity in other muscles, move period activity in a given muscle remained a much better predictor of the change in hold period activity in that same muscle ($R^2 = 0.42 \pm 0.031$, mean ± SEM).

In summary, for a constant target location, on trials in which a muscle moved the finger with greater activity, it also produced greater activity during the subsequent hold period. About 45% of the trial-to-trial variation in the change in hold period activity could be associated with the integral of the preceding move period activity.

## Change in reach period commands alters hold period commands

These EMG patterns illustrated a correlation between move and hold period commands, but did not test whether there was a causal link between the two. That is, trial-to-trial coupling between move and hold period commands arose from variability that was internally generated by the animal. To rigorously test if hold period commands directly followed from move period commands, we next imposed external changes on move period commands and measured if hold period commands changed in a manner consistent with integration. To do this, we instructed participants to reach to a target while holding the handle of a robotic arm (*Figure 3A*), and adapted their move period commands by imposing a velocity-dependent force field (*Albert and Shadmehr, 2016*).

With the force field our goal was to gradually bias the move period commands through the process of adaptation. If hold period commands partly depended on the preceding move period, then biases in move commands should lead to biases in hold commands, even though (1) the hold period commands were never perturbed, and (2) the hold location remained constant. To ensure that hold period commands were never perturbed, on all force field trials the hand was placed in a 'well' that held the hand still at the end of the movement for at least 1.5 s (*Figure 3A*, target-hold well). Next, we used a channel to prevent the hand from suddenly moving off the target while participants transitioned from the target-hold well to the next reaching movement (not shown in *Figure 3A*; see 'partial channel' in Materials and methods).

Unlike the monkey experiments, we did not record EMG during these experiments. Instead, we measured changes in the forces that participants exerted against the handle. These forces served as a low-dimensional proxy for the motor commands sent to the arm muscles. Critically, we drove the adaptation process with forces that acted perpendicular to the trajectory of the hand. Because the learning axis was orthogonal to the primary movement, we could cleanly isolate the component of the motor commands that varied in response to the field, from the motor commands responsible for the primary movement. To measure the forces perpendicular to the primary movement, on some trials the robot produced a stiff channel that connected the start position to the target via a straight line (*Figure 3A*, channel trial). To test the integration hypothesis, we recorded forces perpendicular to the direction of the target during the reach and hold periods, and asked if they were related through the following integration function (i.e. the force analogue of *Equation 1*):

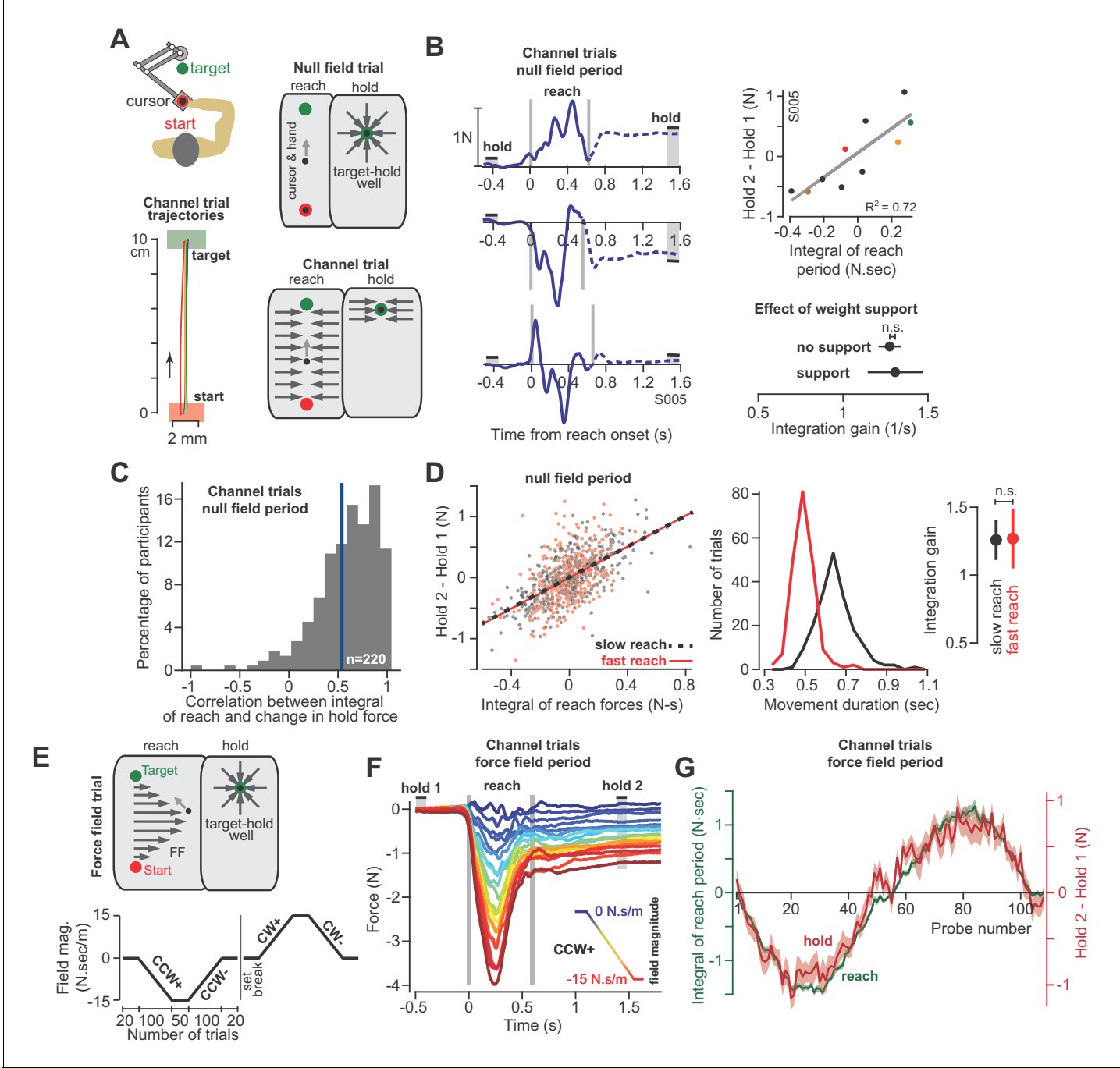

**Figure 3.** The integral of reaching forces correlates with forces produced during the subsequent hold period. (A) Human participants held the handle of a robotic arm and made point-to-point reaching movements (top). On most trials, participants reached freely to the target. After the reach ended, a target-hold well held the hand in place (null field trial). On some trials, hand trajectory was constrained to a straight line (channel trial, trajectories shown at left). (B) Example lateral force traces on channel trials during the null field period (in a single subject). The bars hold 1 and hold 2 indicate the periods in which holding force levels were quantified. At top-right, we show the correlation between the integral of reach period forces and the change in holding force for this representative subject. At bottom-right, we compare the gain of integration (slope of the line at top-right) between conditions with and without weight support for the arm. (C) We calculated the correlation coefficient between the time-integral of reach forces and holding forces across all channel trials in the null field, within each individual (n = 220). The vertical blue line denotes the mean of the distribution. (D) For each subject, we selected their two fastest and two slowest movements in the null field, resulting in two distributions, with each subject represented equally in each distribution. We then performed linear regression on each distribution separately. Error bars for the integration gain are 95% CI at right. At middle, we show the distribution of reach duration. (E) After the null field period, we gradually introduced a velocity-dependent force field (top). We measured moving and holding forces as subjects adapted and de-adapted to counterclockwise (CCW) and clockwise (CW) force fields (bottom). (F) Each trace

*Figure 3 continued on next page*

*Figure 3 continued*

represents the force on one channel trial, averaged across participants during the CCW force field adaptation period. The vertical gray bars denote the start and end of the reaching movement. The color of each traces indicates the force field magnitude at each point in the experiment. The hold 1 and hold 2 bars indicate periods over which holding forces were quantified. (G) On each trial, we calculated the time-integral of forces during reaching (green) and compared these to the change in holding force (red). Values are mean ± SEM across all participants. Statistics: n.s. p>0.05.

The online version of this article includes the following source data and figure supplement(s) for figure 3:

**Source data 1.** The EXCEL source data file contains holding forces and the integral of moving forces as shown in *Figure 3*.
**Figure supplement 1.** Holding forces are sustained across long time intervals.

$$F(h_2) - F(h_1) = k \int_0^T [F(t) - F(h_1)]dt + a \qquad (2)$$

At the start of each experiment, participants (n = 220 in total) reached to the target in a neutral (i.e. null field) condition in which the robot did not produce any forces on the hand (*Figure 3A*, null field trial). Even in the null field period, we observed significant trial-by-trial variability in the perpendicular reach period forces, as shown for a representative subject in *Figure 3B*. During the reach period, the hand pushed slightly to the right on some trials, left on others, or exhibited a bimodal profile. However, because the movements were guided within a channel, the hand followed a straight line that ended at the center of the target (*Figure 3A*, bottom left). Notably, following conclusion of the reach, we observed that the arm generated forces during the hold period (period $h_2$, *Figure 3B*) that were often different than baseline (period $h_1$). Indeed, the change in the hold period forces $F(h_2) - F(h_1)$ was well predicted by *Equation (2)*, as illustrated by data for the representative subject in *Figure 3B* (regression at right), and the entire population in *Figure 3C*. On a within-trial basis, the integral of move period force accounted for 39 ± 2% (mean ± SEM) of the variance in hold period force. Thus, just as EMG patterns exhibited a within-trial relationship between the reach and hold periods, so did the force patterns.

Like the EMG patterns, the force patterns did not appear to be trivially related to biomechanical constraints imposed on the arm due to gravity: the gain of the integration function was the same whether or not the weight of the arm was supported by a frictionless air-sled (*Figure 3B*, two-sample t-test, p=0.90). Remarkably, the relationship between move and hold period forces remained unchanged when we divided the null period reaches of each subject into fast and slow movements (*Figure 3D*, ANCOVA, movement type by move force integral interaction effect on hold force, F = 0.007, p=0.935). Thus, in the null field, forces during the reach and subsequent hold periods showed natural variability. However, on a within-trial basis, the integral of the move period forces appeared to influence the subsequent hold period forces.

At the conclusion of the null field period, we gradually imposed a velocity-dependent force field during the reach (*Figure 3E*). The velocity-dependent forces perturbed the hand perpendicular to the reach trajectory, thus leading to adaptation of reach period forces (*Figure 3F*). The gradual onset of the perturbation produced gradual changes in reach period forces (while also minimizing positional errors throughout the trajectory). Remarkably, as the reach period force changed from one trial to the next, so did the hold period force (*Figure 3F and G*). Notably, the hold period forces were not transient, but sustained, persisting up to 6 s during the entire hold period interval (*Figure 3—figure supplement 1*). The relationship between the change in hold period force and the (now externally-driven) reach period force was again consistent with integration: over the course of adaptation, *Equation (2)* accounted for 48 ± 3% (mean ± SEM across 32 participants) of the variance in the change in hold period forces.

As an alternative hypothesis, we considered the possibility that hold period forces may be a trivial continuation of the forces produced near the end of the preceding reach, not an integration of the entire history of the reach period. To test this idea, we conducted a pair of experiments. In the first experiment (*Figure 4A*), participants (n = 11) reached in a force field that was active only during the second half of the reach. In Phase 1 of the experiment, trial after trial we gradually increased the magnitude of the force field (*Figure 4A*, Phase 1). As expected, participants produced hold period forces that increased with the integral of the reach period forces. In Phase 2, we maintained the force field at a constant magnitude for hundreds of additional trials. The change in hold period

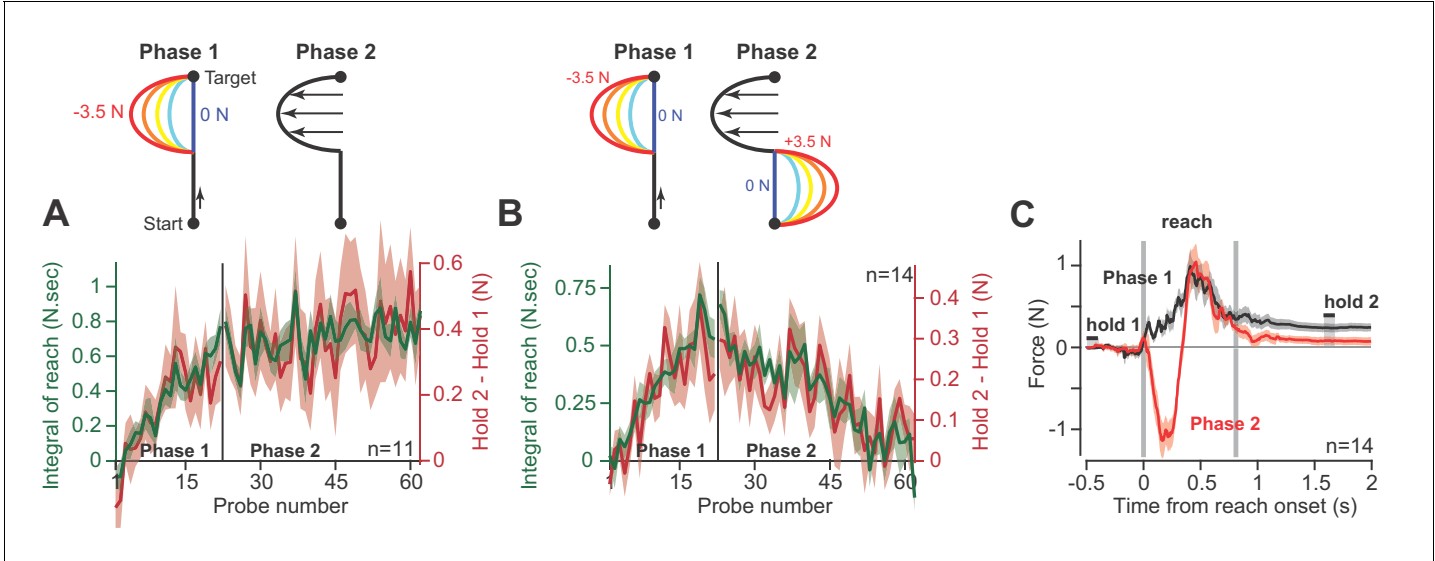

**Figure 4.** Holding forces are an integration of moving forces, not a continuation. We designed a set of experiments to test the possibility that hold period forces are a continuation, not an integral, of moving forces. (**A**) Participants (n = 11) reached in a force field that was active only during the second half of the reach. In Phase 1 (left), we gradually increased the magnitude of the force field. In Phase 2 (right), we maintained this force field for several hundred trials. We measured the change in holding force (right, red) and the integral of moving force (left, green) throughout adaptation. (**B**) A new set of participants (n = 14) repeated Phase 1 (left), but during Phase 2 (right) an opposite force field was gradually applied to the first half of the movement. As the integral of move period force approached zero in Phase 2, so did holding force. (**C**) The mean force profile over trials sampled from Phases 1 and 2 for the experiment in part B. Values are mean ± SEM across all participants.

The online version of this article includes the following source data for figure 4:

**Source data 1.** The EXCEL source data file contains holding forces and the integral of moving forces as shown in *Figure 4*.

forces and the integral of reach period forces remained correlated during all trials (*Figure 4A*, Phase 2).

In the next experiment (*Figure 4B*), we again exposed participants to a force field that was active only during the second half of the movement. But then, in Phase 2, we gradually added a second force field that was active during the first half of the movement, but in the opposite direction (*Figure 4B*, Phase 2). In this way, the reach period forces should integrate to approximately zero. If holding forces were a simple continuation of the reach period forces, then the addition of the force field during the first half of the movement should not alter the hold forces at the end of the movement. However, this is not what we observed: as the integral of reach period forces approached zero, hold period forces gradually vanished (*Figure 4B*, Phase 2). That is, even though reach period forces were matched just before the end of movement (*Figure 4C*), the ensuing hold period force depended on the entire history of the reach. In addition, we observed no difference in the integration function between each phase of the experiment (paired t-test on slope, p=0.22, paired t-test on intercept, p=0.09).

Together, these observations demonstrated that on a within-trial basis, as the reach period forces changed, so did the ensuing hold period forces. The change in hold period force was partially described via a function that integrated in time the temporal history of the preceding reach period forces.

## Correlations between move and hold commands are only weakly influenced by initial hold activity

*Equations (1) and (2)* use initial hold activity ($u(h_1)$ or $F(h_1)$) both to calculate the change in hold activity, and to estimate move period activity (Materials and methods Section A). We were concerned that some or all of the observed correlations may be due to this common factor that appears on both sides of the equation, and not the integral of the move period activity.

To consider this problem, we noted that for outwards reaching movements in *Figure 1*, this was not a concern because the trial-averaged EMGs were nearly identical before movement onset at the

center location, but differed greatly during the movement to various directions. To address this concern for our data in *Figures 2* and *3*, we re-analyzed the correlation between move and hold activity, but only on trials in which hold 1 activity fell within one standard deviation of the mean. This criterion reduced the variance in hold 1 activity by 75% for our finger movement dataset, and 88.6% for our reach force dataset. Despite this dramatic reduction in hold 1 variability, we found little effect on the measured correlations: the correlation coefficient between integral of move activity and change in hold activity remained 58.4% for muscles in the finger (compared to 64.2% if all trials are included) and 48.5% for reach forces on channel trials in the null field (compared to 52.2% if all trials are included).

In summary, the observed trial-by-trial correlations between move activity and change in hold activity were almost entirely driven by the integration of the move commands, with little dependence on variability in the initial hold activity.

## The postural field during the hold period

Thus far, we have described the state of the limb during the hold period in terms of muscle activity or force generation. However, in order to hold the limb still, the postural controller does not simply produce a force, rather it generates a converging field of position-dependent forces (*Mussa-Ivaldi et al., 1985*; *Shadmehr et al., 1993*). We next asked if this postural field also relied on the commands generated during the preceding reach.

We designed a new experiment in which we measured the postural field following completion of a reach. Participants (n = 27) reached to a target as before, but now, during the hold period they were engaged in a short-term memory task (2-back, *Figure 5A*). As they performed the memory task, the robot slowly displaced their arm in a random direction. In response to the displacement, the postural controller produced restoring forces against the handle, thus allowing us to measure the postural field (*Figure 5B*).

As expected, the postural field's null position was near the target (*Figure 5B*, null point of postural field). However, after participants were exposed to a force field, the postural field changed: the null position was no longer aligned with the target (*Figure 5B*, right). Rather, it shifted by approximately 1 cm (*Figure 5D*; paired t-test, $p<10^{-4}$) in the direction of the force produced during the reach. In contrast, the orientation (*Figure 5D*, paired t-test, p=0.84) and stiffness (*Figure 5D*, paired t-test, p=0.62) of the postural field remained unchanged.

After the reach had ended and the cursor was at the target, we slowly displaced the hand toward the postural null position. We observed that the hold period forces gradually approached zero, and then switched direction and grew larger as the hand was displaced beyond the null position (*Figure 5E*). The holding force at the hand scaled linearly with the distance between hand position and the postural null position (*Figure 5F*). This implied that the hold period forces we had measured in previous experiments (*Figures 3–4*) were a proxy for the location of the null position of the postural controller: the larger the hold period force at the target, the farther the null position of the postural field.

Thus, as the reach period forces changed, so did postural control: the null position of the postural field shifted in the same direction as the change in the preceding move period forces.

## Adaptation of the integration gain

These results create a puzzling scenario. In the presence of a velocity-dependent force field, the reach controller readily adapts and changes the move period forces. However, changes to the move period forces are integrated and cause the hold system to program an entirely different null position. This implies that postural stability will be compromised in the face of an adapting reach controller. To solve this problem, the integrator must also be adaptive: the integration function must change when there is an error between its current output and the desired movement endpoint (*Figure 6A*). Presumably, this error-based adaptation would be driven by unexpected deviations from the hand's desired trajectory as the reach period ends and the hand arrives at the target location.

To test this idea, we examined reach trajectories of individual subjects as they gradually adapted to velocity-dependent force fields. These trajectories exhibited two primary types of errors. The first error happened midway through the movement and was caused by incomplete compensation for

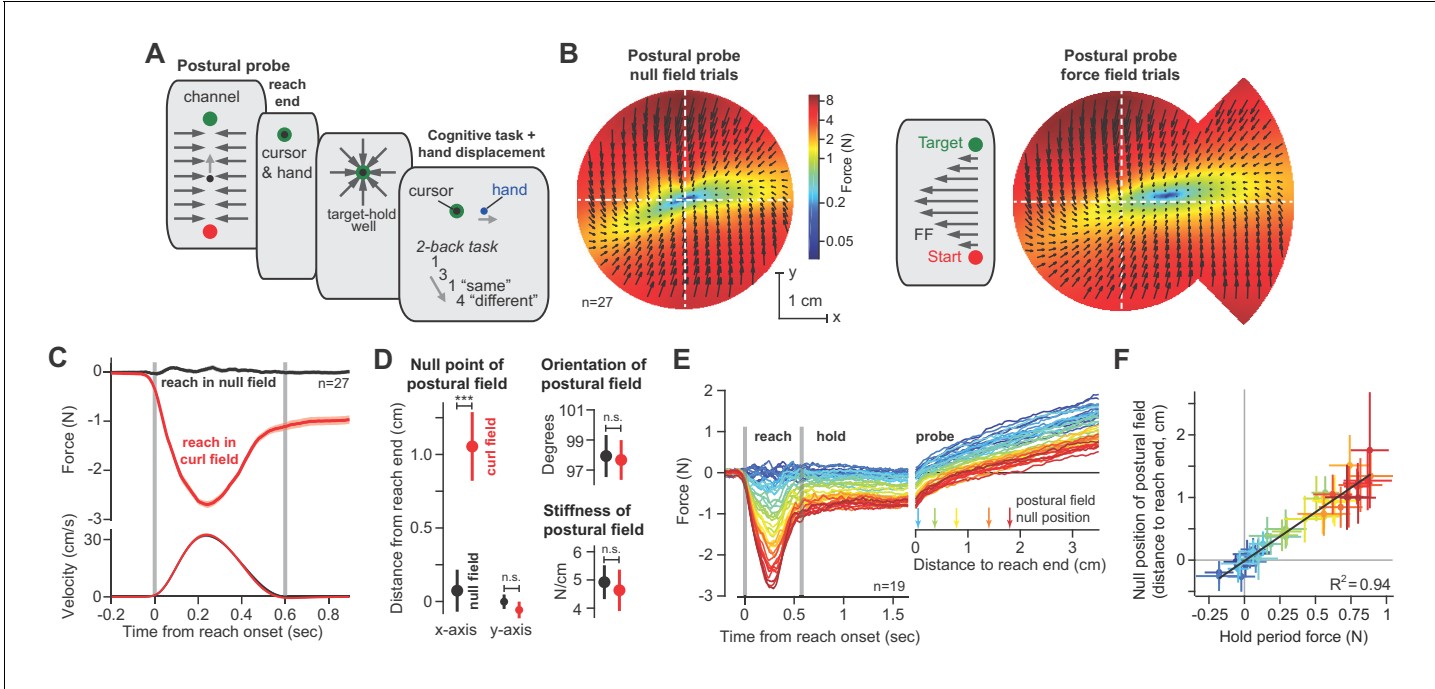

**Figure 5.** The null point of the postural field is set by the integral of reaching forces. (**A**) To measure the arm's postural field, we slowly displaced the hand during the holding period, while participants were distracted with a working memory task. (**B**) We measured the forces applied to the handle (left). We re-measured these forces after participants were exposed to a velocity-dependent curl field (at right). Forces were measured by displacing the hand outwards along 12 different lines. Interior estimates for the force were made using two-dimensional linear interpolation. The magnitude and direction of these interpolated forces are indicated by the black arrows. Color reiterates the restoring force magnitude. The holding position at reach end is located at the intersection of the two dashed white lines. (**C**) We measured lateral forces applied to the channel walls during reaching movements (null field period, black, and curl field period, red). (**D**) We used a two-dimensional spring model to quantify postural field properties: null point, orientation, and stiffness (null field and curl field in black and red). (**E**) To test if holding forces were related to the null point of the postural field, participants (n = 19) were exposed to a curl field that gradually increased over trials. During holding, we recorded hand forces (right inset) as the arm was displaced in the direction of holding forces. Arrows show the location of the null point (zero-crossing) on selected trials. (**F**) We calculated the holding force before displacement of the hand, and the corresponding postural null point on each trial. Values are trial means and 95% CIs for distributions bootstrapped across participants. Linear regression was performed on the bootstrapped estimates (black line). Error bars denote mean ± SEM in panels (**C** and **D**). Statistics: ***$p < 10^{-3}$ and n.s. $p > 0.05$.

The online version of this article includes the following source data for figure 5:

**Source data 1.** The EXCEL source data file contains parameters for our postural field model, as shown in *Figure 5D*.

the velocity-dependent perturbation (*Figure 6B*, the large negative mid-movement error). The second error happened near the end of the movement and was oriented in the opposite direction (*Figure 6B*, 'endpoint correction'). This near endpoint error possessed two properties that were well-suited for integrator adaptation: (1) they occurred late in the movement as the participant attempted to stop their hand within the target, and (2) they were oriented in the direction opposite the shift in postural null point reported in *Figure 5*. To quantify the size of these endpoint errors, we measured the largest 'positive' deviation (or 'negative' if the lateral deviation was in the opposite direction) from the terminal hand position, after the hand exceeded 80% of its reach displacement.

To determine if endpoint errors caused integrator adaptation, we compared the size of these errors during the adaptation process, to the gain of integration observed at the end of adaptation. The size of endpoint error was heterogeneous across our subjects; some participants exhibited large endpoint errors (*Figure 6B*, S11) while others exhibited small endpoint errors (*Figure 6B*, S5). Critically, we found that participants with larger endpoint errors ultimately produced smaller holding forces (*Figure 6C*). In other words, these errors appeared to reduce the gain of integration. In fact, about 40% of the variability in integration gain could be explained by the magnitude of the late endpoint errors (*Figure 6D*).

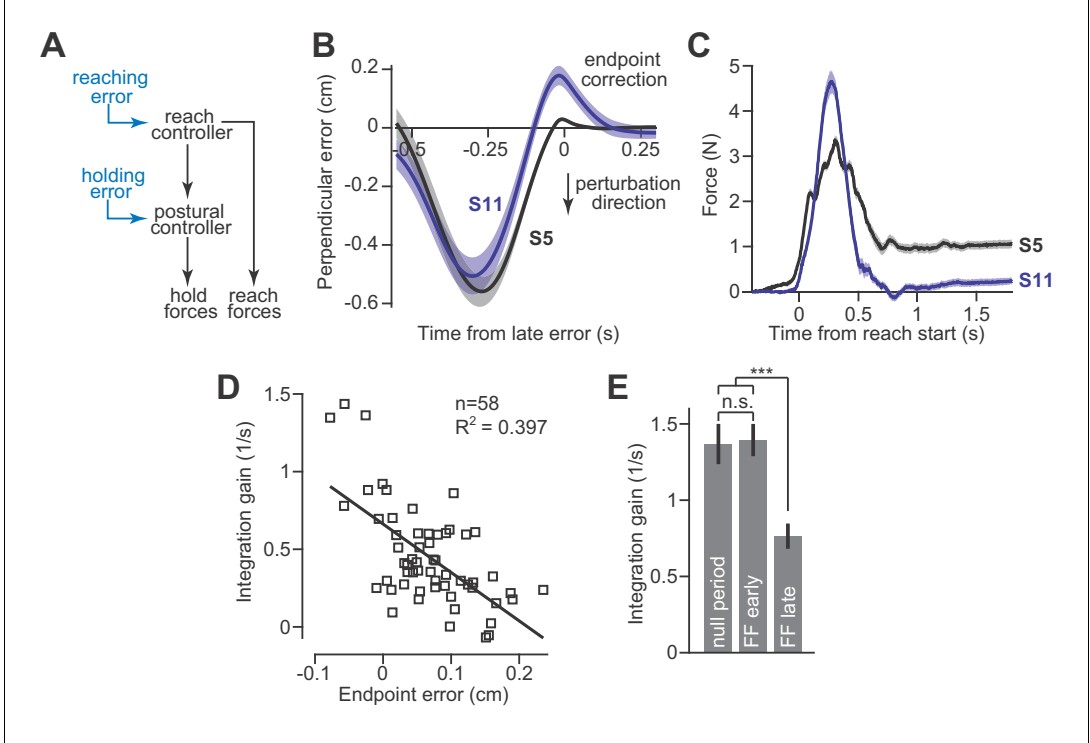

**Figure 6.** Adaptation of the integration gain. (A) To maintain endpoint stability after adaptation of the reach controller, the postural controller must also adapt. (B) We hypothesized that integrator adaptation would be driven by errors in hand trajectory that occur near the end of the reach. To detect these errors, we looked for deviations in the reach trajectory after the reach exceeded 80% of its displacement. We spatially aligned reaches by subtracting off the terminal hand position and then temporally aligned the reach trajectories to the point in time at which the hand had the largest endpoint error. These errors are marked as 'endpoint correction'. Here, we show the average reach trajectory during adaptation to a gradual force field for two subjects, one who exhibited integration (S5), and one who did not (S11). (C) The average forces produced at the end of adaptation for the same two subjects. Note that the subject with large errors near reach endpoint no longer generated holding forces at the end of adaptation. (D) We measured the gain of integration at the end of the adaptation period, for subjects that adapted to gradual force fields. We also measured the magnitude and sign of the late reach errors. Larger errors led to a reduction in gain. Each data point is one subject. (E) To confirm that the integration gain changed over the course of adaptation, and not immediately upon introduction of the force field, we compared the gain during the null period, with gains measured during early and late parts of adaptation using a repeated measures ANOVA. Values are mean ± SEM across participants. Statistics: ***$p < 10^{-3}$ and n.s. $p > 0.05$.

The online version of this article includes the following source data for figure 6:

**Source data 1.** The EXCEL source data file contains data for panels D and E of *Figure 6*.

This adaptation of integration gain progressed over time (*Figure 6E*, repeated measures ANOVA, F = 12.60, p<0.001). In the early part of training, the integration gain was no different than that of the pre-adaptation null trials (*Figure 6E*, post-hoc comparison, p=0.88). However, late in training, the integration gain had decreased substantially (*Figure 6E*, post-hoc tests, p<0.001 for comparison of late adaptation with both null field trials as well as early adaptation trials).

These data suggest that *Equation (2)* alone cannot predict the change in holding forces. In certain conditions, for example velocity-dependent force fields, the reach period commands change, but if one is to hold the hand at the target, then the hold period commands cannot simply integrate the preceding move period commands. Rather, as the move period commands adapt, so too must the integration function. Like the adaptation of movement commands, the adaptation of hold commands does not occur instantaneously, but appears to emerge gradually as errors near the end of the movement are experienced repeatedly. This process may also explain why holding forces gradually diminish during adaptation to an abrupt force field, where endpoint errors are large (*Sing et al., 2009*).

# Differential contributions of the corticospinal tract to reaching and holding

The CST conveys the cortically generated reach commands to downstream motor structures. Does this same pathway also convey postural signals, or does a separate, downstream structure receive and then integrate the reach commands? If both reaching and holding commands are conveyed in the CST, then damage to the CST above the level of the brainstem should disrupt both the generation of forces during reaching, and its integration during the hold period. However, if the integrator is downstream to this level, then damage to the CST might result in deficient reach commands, but spare the process of integration, resulting in hold commands that reflect the within-trial integration of the now deficient reach commands.

To examine these possibilities, we recruited stroke patients (n = 14) who had suffered lesions affecting the CST pathway from the cortex through the internal capsule (*Supplementary file 1*). The patients exhibited profound impairments, as demonstrated by difficulty with extension of their arm during unsupported reaching (*Zackowski et al., 2004*; *Roh et al., 2015*; *Figure 7A*, patient S015). To improve their reach capacity, we supported the weight of their arm in the horizontal plane (frictionless air sled), which allowed them to better extend their arm at the elbow, enabling them to make planar, point-to-point reaching movements while holding the handle of the robot arm (as in *Figure 3A*).

As has been noted before (*Scheidt and Stoeckmann, 2007*; *Coderre et al., 2010*), movements of the paretic arm exhibited erratic trajectories (*Figure 7B*), increased movement duration (paretic vs. control) of approximately 41% (paretic vs. non-paretic, paired t-test, p<0.01; paretic vs. control, two-sample t-test, $p<10^{-4}$), and reach endpoints that terminated nearly 89% (paretic vs. control) further away from the target location (paretic vs. non-paretic, paired t-test, p<0.01; paretic vs. control, two-sample t-test, p<0.001). The reaching impairment coincided with a marked increase in the trial-to-trial variability of move period forces (traces in *Figure 7C* solid lines; *Figure 7D* paretic vs. control, Wilcoxon rank-sum, p<0.001; *Figure 7D* paretic vs. non-paretic, Wilcoxon signed-rank, p=0.058). However, like healthy subjects, these move period forces, no matter how variable, terminated with stable holding forces (traces in *Figure 7C* dashed lines). The trial-by-trial variability of the hold period forces, like the move period forces that preceded them, was significantly greater in the paretic arm of the patients (*Figure 7E*; paretic vs. control, two-sample t-test, p=0.01; paretic vs. non-paretic, paired t-test, p=0.06).

If move period commands were integrated into hold period commands, the increased variability in holding forces (*Figure 7E*) could arise indirectly from the normal integration of the highly variable moving commands. If this were true, the variability in the moving and holding commands would be similarly structured in both healthy subjects and stroke patients. To test this idea, we quantified the within-trial correlation between change in hold period forces and the integral of the preceding move period forces (*Figure 7F*, left column, representative subjects). Remarkably, in the null field trials, the coupling between move and hold periods was intact in stroke patients (*Figure 7H*, paretic vs. control, two-sample t-test, p=0.14; paretic vs. non-paretic, paired t-test, p=0.63).

We next used adaptation to systematically manipulate move period forces. Because force field adaptation is largely a cerebellar-dependent process (*Smith and Shadmehr, 2005*), despite damage to the CST the patients learned to alter their move period forces. As the move period force changed in the paretic arm, so did the change in hold period force (*Figure 7F*, right column, example subjects). Once again *Equation (2)* provided a reasonable account of the within-trial relationship between the move period and the change in hold period forces for the paretic arm, non-paretic arm, and the dominant arm of age-matched control subjects (paretic vs. control, two-sample t-test, p=0.08; *Figure 7I*: paretic vs. non-paretic, paired t-test, p=0.24). Notably, the integration gain was not significantly different across the paretic and non-paretic limbs of the patients, nor across the patients and age-matched controls (*Figure 7J*; paretic vs. control, two-sample t-test, p=0.86; paretic vs. non-paretic, paired t-test, p=0.91). In other words, the integration function was similar in healthy participants and stroke patients.

In summary, damage to the CST severely affected the reach period commands, resulting in high trial-to-trial variability. However, the change in hold period commands remained coupled to the integral of the preceding reach commands in both null field and force field trials. The gain of this integration in the stroke patients was not different than that of healthy controls, suggesting that CST

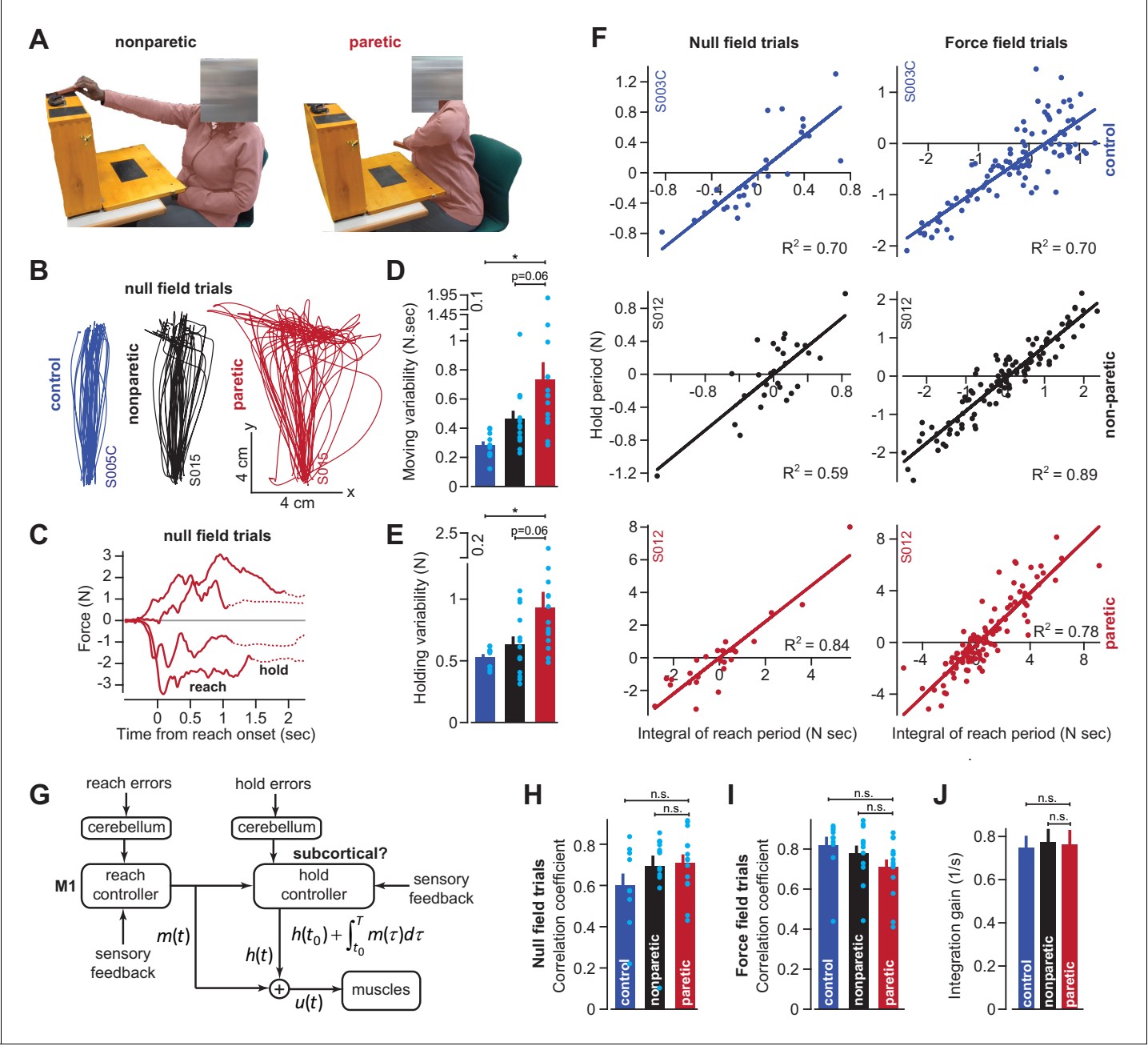

**Figure 7.** Cortical reaching commands are integrated in a subcortical area. (A) Stroke survivors (n = 14) participated in a set of clinical exams to measure functional impairment. Shown are isolated images for an extension-based task, for the non-paretic (top) and paretic (bottom) arms of an example participant (S015). The instruction is to place a rectangular block on the elevated surface. Images show the moment of maximal extension for the paretic (right) and nonparetic (left) arms. (B) To improve the range of motion of the arm, patients and healthy controls performed reaching movements holding the robotic handle, with the arm supported by an air sled. Shown are example trajectories during an initial null field period for the representative patient (black is nonparetic, red is paretic) in A, and a control participant (blue). (C) Example force traces during null field block in channel trials. The solid line denotes forces during moving. The dashed line denotes forces during holding still. (D and E) We measured the integral of moving forces (D) and holding forces (E) on each channel trial. We measured the trial-by-trial variability (standard deviation) of these quantities across all movements in the null field. (F, H, I, and J) We compared trial-by-trial fluctuations in moving and holding forces during the null field period (F, left panel). Next, we gradually adapted subjects to a velocity-dependent force field and compared within-trial integral of moving forces with subsequent holding force (F, right panel). Data are shown for a representative stroke patient and healthy control. We calculated the correlation coefficient between reaching and holding forces during the initial null field period (H) and force field period (I). We measured the slope of the integration function (i.e., the integration gain) across all trials within individual subjects (J). (G) Our conjecture that the cortex generates reaching commands which are then
*Figure 7 continued on next page*

*Figure 7 continued*

integrated in a subcortical area spared by cortical stroke. Values are mean ± SEM across participants. Points represent individual trials in **F**. Points represent individual subjects in **D**, **E**, **H**, and **I**. Statistics: *p<0.05, and n.s. p>0.05.

The online version of this article includes the following source data for figure 7:

**Source data 1.** The EXCEL source data file contains holding forces and the integral of moving forces as shown in *Figure 7*.

damage impaired the reach commands, but not the process of integration that may have generated the hold commands.

## Discussion

Current computational models of reaching view move and hold periods as events that take place in sequence: the sensory representation of the target engages a feedback controller that moves the arm, and then once the desired endpoint has been achieved, a separate controller is engaged that produces the sustained commands that hold the arm (*Yadav and Sainburg, 2011*; *Lametti et al., 2007*; *Todorov and Jordan, 2002*; *Ghez et al., 2007*). These models have usually assumed that the motor cortex is responsible for generating the move period as well as the ensuing hold period commands (*Humphrey and Reed, 1983*; *Kurtzer et al., 2005*). Here, our experiments suggest the possibility of a different architecture (*Figure 7G*), one in which movement commands are integrated in real-time (*Figure 1B*) by a separate network of neurons of possibly subcortical origin, resulting in holding commands. If true, this would imply that control of the arm shares a design principle present in control of the eye (*Cohen and Komatsuzaki, 1972*; *Cannon and Robinson, 1987*; *Crawford et al., 1991*; *McFarland and Fuchs, 1992*; *Miri et al., 2011*; *Shadmehr, 2017*; *Godaux et al., 1993*; *Cheron et al., 1986*) and the head (*Crawford et al., 1991*; *Klier et al., 2002*; *Shaikh et al., 2013*).

We measured muscle activity during point-to-point reaching in the vertical plane and found that across directions and durations, a form of mathematical integration related muscle activity during the hold period with the preceding reach period activity (*Figure 1*). When the start and end positions of finger movements were kept nearly constant, there was still large variability in the hold period EMG of many finger muscles. On a within-trial basis, for all muscles recoded the integral of the move period EMG partly accounted for the change in hold period EMG. In contrast, activity before the start of the movement, co-contraction, or even finger position itself were poor predictors of the final hold period EMG. Thus, fluctuations in the integral of the move period commands influenced the change in hold period commands.

Next, we altered the move period commands and asked whether that change altered the commands that were generated during the hold period. We approached this question in healthy participants, as well as patients who had survived a stroke affecting their CST above the brainstem. In both populations, as reach period forces changed during force field adaptation, so did the subsequent hold period forces, as predicted by integration (*Figures 3* and *7*). Integration was also observed during the null period prior to the introduction of the force field, effectively ruling out the possibility that moving and holding correlations arose due to reach adaptation. Critically, the same integration function was observed for both healthy participants and stroke patients, suggesting that the putative integrator might reside within a subcortical structure.

As an alternative to integration, we considered the possibility that the hold period forces may be a continuation of the reach period forces, not an integration of the entire period. To test this idea, we changed the reach period forces via adaptation to a bidirectional force field (*Figure 4*), one in which the integral of the move period forces was zero, but not the forces near the start or end of the move period. We found that as the integral of the move period force approached zero, so did the change in hold period force.

Finally, we considered the fact that in order to hold the arm at a specific location, the nervous system must produce a postural field (*Mussa-Ivaldi et al., 1985*; *Shadmehr et al., 1993*; *Giszter et al., 1993*). We measured this field by engaging subjects with a working memory task, while slowly moving their hand away from the target location. We found that as the reach period commands changed, so did the null position of the postural field. The magnitude of this shift was proportional to the integral of the preceding reach forces. These results suggested that the hold forces we

measured in our adaptation experiments were generated by a disparity between hand position and the null point established by the integration of moving commands. This mismatch may help explain the paradoxical illusions in perception of arm position (*Mattar et al., 2011*; *Darainy et al., 2013*) that accompany force field adaptation.

The idea that move period activity is integrated into a specific null point might explain why we observed poor trial-by-trial correlation between hold activity and finger position in *Figure 2*. Note that the tube housing the finger constrained the motion of the finger to an arc, and thus may have prevented it from moving to its true 'null position' in three-dimensional space. Therefore, if move commands integrated to a position outside of the tube, for a fixed tube rotation, there could be many null positions lateral to the tube, thus decreasing the observed correlation between the rotational position we measured and the associated EMG. In other words, trial-to-trial variability in move period activity should integrate to different magnitudes of hold activity, thus leading to hold positions that do not necessary lie within the tube.

The notion that holding commands control a null position is reminiscent of earlier theories in which the principal role of the motor system was to specify equilibrium positions for the arm (*Feldman, 1966*; *Feldman, 1986*). These theories posited that movement arose from the transition between holding locations. In sharp contrast, our results imply that the hold commands are generated in real-time via integration of the ongoing move period commands.

## A hypothetical architecture for control of arm posture

The model presented in *Figure 7G* represents our conjecture regarding architecture of the reach and hold controllers. In this conjecture, movements are encoded by the cortex, resulting in the move signal, termed $m(t)$, which is integrated in real-time by an unknown subcortical area. The integral of move commands is then added to initial hold activity, yielding a feedforward estimate of the commands required for holding still, termed $h(t)$. During a movement, the commands that arrive at the motoneurons are a sum of commands for moving and holding, $u(t)$.

There are a number of predictions that arise from this model. Motor commands required to move the arm to a target position are not fixed, but vary because of interactions with external objects, interaction torques that arise when the body is in motion, and over time as our bodies change. We know that the move system continuously adapts to these novel dynamics. A similar process of adaptation would also be required of the proposed reach integrator.

Indeed, in the oculomotor system, move period and hold period commands can both undergo adaptation, and this adaptation depends on the cerebellum (*Shadmehr, 2017*). However, different regions of the cerebellum are required for adaptation of the move and hold periods (*Shadmehr, 2017*). *Optican and Miles (1985)* demonstrated that the oculomotor integrator could be adapted by translating the target position as the eye transitioned from a saccade to gaze holding. In our reaching experiments, we found evidence for a similar adaptation mechanism. Errors near the end of the reach that were consistently encountered as the arm attempted to stop within the target location reduced the gain of the integration function (*Figure 6*). Through adaptation of the hold system, the arm would be able to cope with novel dynamics during the reach while also maintaining the ability to hold the arm at the target.

While this adaptation mechanism would achieve endpoint stability over the sequence of many movements, immediate corrections in the integrator output would be handled through the parallel operation of sensory feedback (*Figure 7G*, sensory feedback). This scheme would resemble a neural integrator for control of the head in the interstitial nucleus of Cajal (*Crawford et al., 1991*; *Klier et al., 2002*; *Shaikh et al., 2013*), which relies on proprioceptive and visual feedback (*Shaikh et al., 2013*). The importance of these feedback pathways is further illustrated by conditions in which the arm is passively moved to a new position.

We think that this new model of reaching might shed light on a number of interesting puzzles. For example, transient inhibition of the motor cortex during a reach results in 'freezing of the arm' at its current posture, and not loss of muscle tone (*Guo et al., 2015*). That is, despite near complete removal of output from the motor cortex during a reach, commands of unknown origin continue to sustain arm position against the force of gravity. Our model suggests that a distinct structure, possibly located in a subcortical area, integrates the cortically-generated reach commands up until the moment of cortical inhibition, and thus maintains posture despite removal of reach commands. This model is also consistent with the observation that cortical stimulation not only moves the limb, but

also produces specific postures in primates (*Graziano et al., 2002*) and rodents (*Harrison et al., 2012*). Moreover, the total displacement of the limb appears to scale with the duration of stimulation, consistent with the idea that displacement is produced due to integration of motor commands over time.

The idea that integration occurs outside of the motor cortex potentially explains why successful decoders of cortical activity are designed to control cursor velocity, as opposed to cursor position (*Sachs et al., 2016*; *Kim et al., 2011*; *Gilja et al., 2012*). To hold the cursor still, these decoders assume that the cortex encodes a zero-velocity command rather than a position-based command. This observation is consistent with the possibility that the cortex is primarily concerned with dynamic quantities, that is changes in the state of the limb, rather than the maintenance of a specific limb state over time. This idea would account for the observation that neurons in the motor cortex modulate their activity when there is a change in force production, but less so when the constant force is maintained over time (*Shalit et al., 2012*; *Georgopoulos et al., 1992*). To maintain a constant force over time, our conjecture states that a subcortical area integrates phasic activity from the cortex, and maintains its output over time. This idea is consistent with evidence that transient stimulation of the brainstem in decerebrate cats produces sustained (timescale of minutes) changes in extensor muscle force (*Mori et al., 1982*).

## Limitations

*Equations (1) and (2)* describe how the integration of move activity may relate to changes in hold commands, but does not specify the hold command at the target. This reflects the reality that move period commands alone will not determine the terminal position of the arm: the initial arm position must also be taken into account. In other words, to hold the limb at the target, the integrator must not only integrate move commands, it must add this integral to the hold period activity that preceded movement (see *Equation (6)* in Materials and methods; illustrated in *Figure 7G*). We do not know if the integrator internally performs this addition, or if a downstream structure handles this adjustment for initial limb position.

Without a biomechanical model of the arm, it is not obvious why the linear functions in *Equations (1) and (2)* robustly predicted the relationship between change in hold activity and the integral of move activity. It may be that as we test movements that are more complex than simple point-to-point reaches, the observed linearity will break down. For example, how would the reach integrator maintain endpoint accuracy when the arm grasps an object, thus altering its mass? Such a scenario would require an increase in muscle activity to move the larger mass, and then hold it still against the forces of gravity, so it may be the case that integration would naturally account for these positive correlations. Furthermore, over successive movements, inaccuracies in the output of the integrator could be reduced through adaptation of the integration gain as in *Figure 6*. Finally, as is the case for the move controller (*Sheahan et al., 2016*; *Heald et al., 2018*; *Howard et al., 2013*), the reach integrator may store object-specific or environment-specific memories and implicitly alter its integration properties when an often-encountered dynamical context is re-experienced. These mechanisms, along with the parallel output of visual and proprioceptive controllers would allow the arm to be stabilized under different inertial conditions.

These possibilities could be tested in the future, by altering the mass of the arm and recording EMG from various arm muscles. With regard to EMG recording, it should be re-emphasized that the reach period forces measured in our human subject experiments were perpendicular to the direction of movement, and thus represented only part of the complete motor command. Because perturbations were applied perpendicular to the reach trajectory, our measure captured the primary dimension of adaptation. In future studies, a more complete picture of the integration process would be provided by recording EMG during adaptation.

It is unlikely that the motor cortex has no role in postural control. At the very least, the monosynaptic projections from corticomotoneurons (*Griffin et al., 2015*) to alpha-motoneurons in the spinal cord are likely to be active during periods of holding still. We do not know if these hold period signals in the cortex arise from recurrent connections with a subcortical integration system, or from a separate position controller.

Understanding the differential contributions of the cortical systems and the putative subcortical integrator is essential to improve our understanding of neurological disorders such as stroke and dystonia (*Sadnicka et al., 2018*). Identifying the locus of the reach integration may help explain why

these patients exhibit abnormal postures at rest. These abnormalities could arise from a lesion to the integrator circuit, or perhaps more provocatively, from the normal integration of chronically abnormal moving commands.

## Materials and methods

### The reach integrator hypothesis

Muscles are engaged during the period of moving as well as the subsequent period of holding still. We hypothesized that for arm and finger muscles, activity during the movement $u(t)$ could be decomposed into contributions from a move controller $m(t)$ and a hold controller $h(t)$:

$$u(t) = m(t) + h(t) \tag{3}$$

If the move and hold controllers are connected in series, like the control system present for the eye and the head, the hold controller produces its output by integrating in real-time the output of the move controller. An example of this is shown in *Figure 1B*, in which EMG of ant. deltoid is decomposed into move and hold commands. Here, we explain this decomposition and derive some of its predictions.

For a movement from position $h_1$ to position $h_2$ the hold controller first produces commands for holding at $h_1$ and then transitions to holding commands at $h_2$ through integration of $m(t)$. Here is an integration function that could accomplish this task:

$$h(t) = u(h_1) + k \int_{t_0}^{t} m(\tau) d\tau \tag{4}$$

In the above equation, $u(h_1)$ represents hold activity at position 1. Combination of *Equations 3 and 4* yields:

$$u(t) = m(t) + u(h_1) + k \int_{t_0}^{t} m(\tau) d\tau \tag{5}$$

Given the measurement $u(t)$, for example EMG from a given muscle, we can decompose the measurement into its constituents $m(t)$ and $h(t)$ by solving the above equation iteratively for $m(t)$. This is what we did to plot the traces in *Figure 1B* (we assumed $k = 1$).

To evaluate the predictions of this equation, we measured movements of duration $T$, between two positions $h_1$ and $h_2$. At the end of the movement, the move commands go to zero yielding:

$$u(h_2) = u(h_1) + k \int_{t_0}^{T} m(\tau) d\tau \tag{6}$$

Unfortunately, we do not have an a priori estimate of the integration gain $k$. Thus, during movement we cannot uniquely estimate $m(t)$ and $h(t)$. To proceed, we made a simplifying assumption: the move commands could be approximated by taking the overall EMG signal $u(t)$ and subtracting off the hold commands measured at the start of movement:

$$u(h_2) \approx u(h_1) + k \int_{t_0}^{T} [u(\tau) - u(h_1)] d\tau + a \tag{7}$$

Here, we added a bias term, $a$, to account for systematic error introduced by our approximation. Rearranging the terms in *Equation (7)* yields the prediction of the hypothesis:

$$u(h_2) - u(h_1) \approx k \int_{t_0}^{T} [u(\tau) - u(h_1)] d\tau + a \tag{8}$$

This states that the change in hold period activity should be roughly a linear function of the integral of muscle activity during the preceding movement with a gain $k$. Thus, from this approximation we arrive at *Equations (1) and (2)* in the main text.

## Monkey experiments

We performed four sets of experiments: (1) reaching movements in non-human primates, (2) finger movements in non-human primates, (3) reaching movements in healthy humans, and (4) reaching movements in stroke patients.

### Reaching movements

Two monkeys participated in a reaching task described elsewhere (*Lara et al., 2018a*; *Lara et al., 2018b*). Briefly, at the start of each trial the monkey held its hand at a central home location. After a 'GO' cue, the monkey reached to one of eight peripheral targets displayed in the vertical plane on a monitor (*Figure 1B*), and then held its arm at the target for at least 0.5 s. After this holding period, the monkey returned its arm to the central home location and held it there for at least 0.5 s, until the start of the next trial.

On each trial, hand position was recorded through infrared optical tracking of a bead fixed to the third and fourth fingers. The activity of several muscles was also recorded using intramuscular electrodes. In Monkey A, these muscles included the anterior, medial, and posterior deltoid, the medial and lateral bicep, and the upper and lower trapezius. From Monkey B, these muscles included the anterior, medial, and posterior deltoid, the pectoralis, the brachialis, the medial and outer bicep, and the upper and lower trapezius. EMG signals were filtered (10–500 Hz), digitized at 1 kHz, rectified, and smoothed with a Gaussian kernel (standard deviation of 20 ms). Muscle activity was then averaged across movements towards each target, separately. We normalized the data by setting to 0 the average muscle activity at the center hold location, and setting to 1 its maximum activity in the task.

We asked whether activity in a given muscle during the hold period could be related to its activity during the preceding reach period (*Equation 1*). For our analysis, the pre-movement hold activity $u(h_1)$ was quantified as the mean activity in the $[-700, -350]$ ms period relative to movement onset for outwards reaches, and $[-300, -200]$ ms period relative to movement onset for return reaches. The post-movement holding activity $u(h_2)$ was quantified as the mean activity over $[+300, +450]$ ms period after movement termination for both outwards and return movements. We started this interval 300 ms after movement offset to allow time for muscle dynamics to settle. And finally, the bounds for integration (i.e. $t_0$ and $T$ in *Equation 1*) were set as 140 ms before movement onset up until movement offset. We started integration prior to movement onset to capture changes in muscle activity that preceded change in kinematics.

Of the 320 different movement types (20 muscles x two reaches per trial x eight targets), six movements (1.9% of trials) had reach durations that were too slow to gain an accurate measurement of holding activity prior to the start of the next movement. To identify these trials, we used a cutoff for movement duration of 850 ms. The six trials with movement durations that exceeded 850 ms were not included in our analysis.

We fit the integration parameters, $k$ and $a$, in *Equation (1)* in the least-squares sense. To determine if the integration gain differed for outward and return movements (*Figure 1G*), we fit the outwards and return movements separately, and then tested for a difference in integration gain with a paired t-test. To determine if integration gain differed for movements of different durations, for each movement type (16 possible movements, eight directions for outwards and return movements), we sorted movements into fast and slow, based on the median movement duration. Because muscles were recorded on different sets of sessions, the average movement durations for each muscle differed. For the slow and fast movement groups, we fit the integration parameters in *Equation (1)*, and tested for a difference in integration gain with a paired t-test.

To determine how gravity impacted the gain of integration, we separated movements into two groups: (1) the horizontal movements to and from Targets 2 and 7 in *Figure 1B*, and (2) the other six movements (all contained a vertical component). Because the horizontal group groups consisted of only four movements (2 out and 2 back) for each muscle, we used a different technique to test for differences in integration gain. We collapsed data across all muscles and movements in the horizontal group, and then separately in the vertical group. We then fit integration parameters in *Equation (1)*. To test if there was a difference in the integration gain (i.e. the slope of the linear regression), we tested for a group by move period interaction effect on the hold period activity, using an ANCOVA.

We also tested if the maximum activity during moving (as opposed to the integral) was an equally good predictor of holding activity. For this, we quantified either the max activity (if the muscle increased activity) or min activity (if the muscle decreased activity) over the entire movement. We then regressed holding activity onto the max or min activity, for each muscle separately. We compared the $R^2$ of this fit to that of the integral fit using a paired t-test (*Figure 1F*, right).

## Finger movements

To examine the within-trial covariance between the commands generated during the movement period and the subsequent holding period, we considered finger movements (*Soteropoulos et al., 2012*). Two monkeys (R and D) used their index finger to track a visual target (*Figure 2A*). The index finger was splinted within a narrow plastic tube, constraining movements to the metacarpophalangeal joint. The hand on the recording arm and all other digits were placed in a padded pocket which prevented movement. The recording arm was placed in a sleeve to prevent movement. The contralateral arm was not restrained.

On each trial, the target moved between two positions at a speed of 12 deg/s. On flexion trials the target moved from 12° to 24°. On extension trials, this order was reversed. The finger flexion movements were resisted with a spring load that measured 0.026 N·m at 12° and 0.048 N·m at 24°. Each trial started with a rapid movement to the start position. The monkey then held its finger at the start location (hold 1) for 1 s, and then made a slow tracking movement to the target over a 1 s interval (e.g. *Figure 2B*). The trial ended with a 1 s hold at the target (hold 2).

We recorded activity of nine muscles of the finger and forearm (1DI, AbPL, EDC, ECR, ECU, FDS, FDP, FCR, and FCU) on each trial and on each monkey (e.g., *Figure 2C*). Muscle activity was measured using subcutaneous electrodes. In order to analyze single-trial activity, we first rectified the data, and then smoothed it with a Gaussian kernel (standard deviation of 200 ms). After this, we normalized the data by setting to 0 the average muscle activity at the 12° hold period, and setting to 1 its maximum activity during movement.

We used *Equation (1)* to test whether there was a within-trial relationship between move period and the subsequent hold period activity of each muscle. We focused on flexion movements in which the finger moved against the external load. For this analysis, pre-movement holding activity $u(h_1)$ was the mean activity over a 400 ms interval starting 1 s prior to movement onset. The post-movement holding activity $u(h_2)$ was the mean activity over a 200 ms period starting 700 ms after movement termination. And finally, the bounds for integration (i.e. $t_0$ and $T$ in *Equation (1)*) were set as 150 ms before movement onset up until movement offset.

We considered data across all muscles and trials (n = 6177 for Monkey R, n = 2830 for Monkey D). We then fit the integration parameters, $k$ and $a$, in *Equation (1)* in the least-squares sense, for each monkey separately. We quantified the variance in holding activity accounted for by the integral of moving activity. For visualization, we also computed the 95% confidence ellipses that describe the joint distribution of holding force and the integral of moving force across all trials (*Figure 2E*).

We compared integration to an alternative hypothesis: fluctuation in the final holding activity was caused by fluctuation in the initial holding activity (*Figure 2D*). This is described by the linear equation $u(h_2) \approx k u(h_1) + a$. To test the alternative hypothesis, we collapsed data across all muscles and trials and regressed hold 2 activity onto hold 1 activity, and computed the variance accounted for (i.e. $R^2$). For visualization, we also computed the confidence ellipses that describe the joint distribution of hold 2 activity and hold 1 activity.

We also considered the possibility that the observed correlations between move and hold period activity could be spuriously generated by the physical constraints required to move and hold against the spring load. That is, larger movements would require more work against the spring, leading to larger displacements that in turn would require greater hold force. This alternative hypothesis hinges on two relationships. One, trial-by-trial changes in hold position must correlate with trial-by-trial changes in hold EMG activity. Two, trial-by-trial changes in the finger displacement (i.e. movement size) must correlate with the integral of moving EMG activity. To test these possibilities, we used linear regression. First, we regressed muscle activity during the target hold period onto the position of the finger during the hold period. In addition, we also regressed the change in hold activity from the start location to the target location, onto the change in finger position across these two periods (i.e. the total displacement of the spring). Finally, we also regressed the integral of move period activity

onto the change in finger position. In all cases, we performed separate regressions for each muscle, and then averaged the resulting $R^2$ values across muscles.

Note that there is a critical difference between regressing hold activity on hold 2 position alone versus regressing hold activity onto the combination of the hold 1 and hold 2 conditions. Regression within, but not across conditions, is the appropriate way to measure the trial-by-trial correlation between EMG and position without the influence of spurious across-condition correlations (*Makin and Orban de Xivry, 2019*). If hold 1 and hold 2 are combined into the same regression (thus doubling the number of data points), these combined measurements account for 42 ± 4% of the variance in EMG activity in individual muscles, as compared to approximately 2% for hold 2 activity alone. The inflated variance accounted for is caused by the large separation between the associated distributions for hold 1 and hold 2 (see *Figure 3C* at right for an example), not by a strong coupling between EMG and position *within* any of these distributions. Therefore, we were careful not to collapse across both hold 1 and hold 2 when quantifying trial-by-trial correlation between EMG and finger position, to avoid these spurious correlations (*Makin and Orban de Xivry, 2019*). With that said, we mention the combination of hold 1 and hold 2, to confirm that the spring load does indeed require a substantial modification in the EMG activity of individual muscles, even though trial-by-trial variations in EMG activity at the hold 2 position are only very weakly related to position (see the main text).

Finally, we considered the possibility that the observed correlations between move and hold period activity could have been caused by trial-to-trial fluctuations in co-contraction. If on some trials, the finger was stiff and co-contracted its muscles, and on other trials less so, we may observe correlations between move and hold activity, without any relation to hold position. To change finger stiffness, agonist and antagonist pairs of muscles would exhibit correlated increases or decreases in activity. In other words, we should be able to predict the hold period activity in one muscle not solely based on its move period activity (as in the integration hypothesis) but also on the activity of simultaneously recorded muscles. To test this idea, we regressed hold period activity of each muscle onto the integral of move period EMG in other muscles. For a given muscle, we performed this regression separately for all of its possible pairs, calculated the $R^2$ value for each pair, and then averaged across pairs to obtain a single $R^2$ value per muscle. We report the average $R^2$ value across all muscles. For context, we compare this $R^2$ value with that obtained by regressing hold period activity in a muscle onto the integral of move period activity in that same muscle.

## Human experiments

All human subject experiments were approved by the Institutional Review Board at the Johns Hopkins School of Medicine. Our healthy human cohort consisted of n = 223 individuals. Healthy participants ranged from 18 to 61 years of age (mean ± SD, 25.2 ± 7.9) and included 128 males and 95 females.

Our stroke patient cohort consisted of n = 14 adults that had suffered damage to the corticospinal tract (CST). The stroke patients ranged from 30 to 80 years of age (mean ± SD, 56.4 ± 15.2) and included six males and eight females. For comparison, we recruited a cohort of healthy age-matched controls who ranged from 28 to 81 years of age (mean ± SD, 60.6 ± 16.3) and included five males and five females. There was no significant difference in age between the patient and older healthy control populations (two-sample t-test, p=0.53).

The stroke patients we recruited had survived a stroke affecting cortical or subcortical white matter associated with the CST. Patients were selected based on MRI or CT scans, and/or available radiologic reports. Scans and/or reports were corroborated to determine the level at which the white matter of the corticospinal tract (CST) was lesioned. *Supplementary file 1* provides the level of the brain at which the white matter was damaged.

We measured the degree of motor impairment in the patients using the Fugl-Meyer Assessment (FMA) and the Action Research Arm Task (ARAT). Two separate raters scored these assessments in each patient. In each limb, we measured the strength of elbow flexion and extension and shoulder horizontal adduction and abduction using a dynamometer (microFet 2). During measurements, participants rested their arm on a side table so the arm rested slightly below shoulder level. Strength measurements were repeated twice, the maximal force was recorded on each effort, and forces were averaged over repetitions. FMA scores, ARAT scores, strength, and other patient

characteristics are reported in *Supplementary file 1*. Missing entries in table indicate that the patient was unable to perform the desired action.

## Overview of human reaching experiments

In all our human experiments (healthy participants and stroke patients), participants held the handle of a planar robotic arm (*Figure 3A*) and made point-to-point reaching movements between targets in the horizontal plane. For stroke patients and age-matched controls, the arm was supported by a frictionless air-sled. In addition, both the paretic (contralateral to lesion) and non-paretic (ipsilateral to lesion) arms were tested. For all other participants, the subject supported the weight of their own arm, and only the dominant arm was tested.

As the subject held the robot handle, the forearm was obscured from view by an opaque screen. An overhead projector displayed a small white cursor (diameter = 3 mm) on the screen that tracked the motion of the hand. Visual feedback of the cursor was provided continuously throughout the entirety of the testing period, except where otherwise noted. Throughout testing, we recorded the position of the robot handle using a differential encoder with submillimeter precision. We also recorded the forces produced on the handle by the subject using a 6-axis force transducer. Data were recorded at 200 Hz. Except where otherwise noted, kinematic time series were aligned to the onset of movement at the time point where hand velocity crossed a threshold of 1 cm/s.

At trial onset, a circular target (diameter = 1 cm) appeared in the workspace, coincident with a tone that cued reach onset. After stopping the hand within the target, a holding period of various durations (1.8 to 6.5 s) ensued where subjects were instructed to continue holding the handle within the target. After this holding period, a random inter-trial-interval sampled uniformly between 0.3 and 0.4 s elapsed prior to the start of the next trial.

At the end of each reach, coincident with the start of the holding period, movement timing feedback was provided. If the reach was too fast (or too slow), the target turned red (or blue) and a low tone was played. If the reach fell within the desired movement interval the target 'exploded' in rings of concentric circles, a pleasing tone was played, and a point was added to a score displayed in the upper-left-hand corner of the workspace. For stroke patients and age-matched controls, the desired movement duration was 600–800 ms. For all other participants, this interval was 450–550 ms. Participants were instructed to obtain as many points as possible throughout the experimental session.

In all human reaching experiments, trials were ordered in pairs of outwards and backwards movements. In other words, each pair started with a reach from a start position to a target (outward reach). The subject then held the arm still at the target position, and then reached back to the start position (the backward reach). Only outwards movements were analyzed here. All backwards movements were performed in a channel, or a partial channel condition (described below).

## Measurement of moving and holding forces in human subjects

At regular intervals throughout each experiment (generally every $5^{th}$ outwards trial) we measured forces in a channel trial (*Scheidt et al., 2000*). On these trials, the motion of the handle was restricted to a linear path connecting the start and target locations (*Figure 3A*). To restrict hand motion to the straight-line channel trajectory, the robot applied perpendicular stiff spring-like forces with damping (stiffness = 6000 N/m, viscosity = 250 N·s/m). This condition maintained the hand in equilibrium along the axis perpendicular to movement. Therefore, the force applied by the robot was equal and opposite to the lateral force applied by the subject, thus serving as a precise measurement of lateral reaching forces. Before analyzing these forces offline, we first subtracted the baseline force from all force time series. We obtained the baseline force by averaging the forces recorded on the channel trials within the null field period at the start of each experiment.

One of the primary objectives of our study was the comparison of moving and holding forces on channel trials. In each of our experiments, the hand remained in the channel during both the moving period and the holding period (a period of time of at least 1.8 s after the reach ended) to allow us to measure both moving and holding forces.

Our primary hypothesis was that the holding forces could be described as an integral of moving forces according to *Equation (2)*. In this equation, $F(t)$ refers to the channel forces exerted by the subject during movement. The quantities $F(h_1)$ and $F(h_2)$ refer to the forces applied by the hand

while holding still at the start position prior to the reach (1) and the target after the reach (2). The parameters $k$ and $a$ refer to the integration gain and offset.

We calculated the initial holding force, $F(h_1)$, as the mean force over a 100 ms period starting 500 ms prior to reach onset. The final holding force, $F(h_2)$, was calculated as the mean holding force over a 100 ms interval starting 900 ms after reach termination. The termination of the reach was determined based on a velocity threshold of 3.5 cm/s. The integral of reach forces was computed over the entire movement duration (from movement onset to movement offset). Movement onset was determined based on a velocity threshold of 1 cm/s.

We conclude this section with a critical point. Holding forces would tend to move the hand off the target during periods of holding still. We hypothesized that these departures might result in adaptation of postural control. For this reason, we wanted to prevent this unwanted motion. On channel trials, we prevented this by keeping the hand in the channel, as described above. However, on all other trials, we also wanted to prevent motion of the hand during the holding process. Therefore, for all outwards movements not performed in the channel (this does not apply to the return movements), we applied a two-dimensional clamp to the hand at the end of the reaching movement (*Figure 3A*). This clamp prevented motion of the hand during the hold period, despite any forces the participant might have applied to the handle, and was programmed as a 'well' within the target location that attracted the hand in two dimensions, with stiff spring-like mechanics (stiffness = 4000 N/m, viscosity = 75 N·s/m). The target-hold well was applied when the hand entered the target location and the hand velocity fell below a threshold value of 3.5 cm/s.

And finally, to make sure that holding forces on outwards movements did not affect the initial motion of the hand on the following backwards movement trial, all return movements were performed in a partial channel. The channel was removed after the hand had traveled 40% of the desired movement amplitude. Therefore, the hand terminated at the start position without any external forces.

## Working memory task

In some of our experiments, we employed a cognitive task to distract participants during measurement of holding forces. The working memory task consisted of a modified 2-back task where subjects were randomly shown an integer between 1 and 4. The integers appeared one at a time so that the next integer replaced the previous integer on the screen (*Figure 5A*). Participants were told to determine if the integer on the display matched the integer shown two numbers in the past. If the integers matched, participants verbally responded with the keyword 'same'. If the integers did not match, participants verbally responded with the keyword 'different'. If the response was correct a pleasant tone was played and a point was added to the experiment score. If the response was incorrect a low pitch tone was played and no point was awarded. To confirm that participants were engaged in the cognitive task, we recorded each correct and incorrect response. Participants were clearly engaged in the cognitive task and responded correctly to 91.8 ± 0.6% of items correctly, at rates of approximately 0.77 ± 0.6 items per second.

## Reaching movements in the null field

A total of 220 healthy subjects participated in these experiments. The general structure of the task is described in the previous two sections. Almost all movements were performed in a null field, that is the subjects freely reached between the start and target positions. At the end of these null field movements, the robot applied a target-hold well during the holding period, as described in a previous section. On some trials, we applied a channel and measured moving and holding forces as described in a previous section. All return movements were performed in a channel, or partial channel as described in a previous section.

This process was the same for healthy subjects and stroke patients. The only differences were that we tested both arms of the stroke patients (paretic then non-paretic), supported the weight of the arm of the stroke patients and their age-matched controls with a frictionless air sled, and allowed for differences in movement timing are described above.

We measured the relationship between the move period forces and the change in holding forces as described in C2. Specifically, we measured the within-trial relationship between these two quantities (*Figure 3B* right and *Figure 3C* for healthy controls; *Figure 7F*, left panel and *Figure 7H* for

stroke patients). At various points throughout the manuscript, we also report the integration gain, or in other words, the slope of the linear relationship between hold forces and the integral of reach forces. We compared the gain of integration for the control group in *Figure 7* to the experiment groups in *Figure 3*, to test if adding weight support altered the integration gain (*Figure 3B*, bottom-right).

It is important to note that we combined several datasets for our analysis in *Figure 3C*. While the structure of each dataset was the same, they differed in the kinematics of the reaching movement. Across these tasks, we varied several parameters of the movement, including the target location (center of the body, to the left of the body, and to the right of the body), the reaching direction (towards the body, away from the body, or at an oblique angle), and the reach amplitude (10 cm or 20 cm). Most subjects reached between the same two locations, but for some subjects, there were two potential targets for each trial. The number of reaching trials performed varied across tasks. They ranged from 40 trials to 288 trials (half were outwards movements; the other half were backwards movements).

In *Figure 3D*, we tested to see if the integration gain differed for fast and slow movements. For each participant, we sorted their channel trial reaching movements in the null field period according to their duration. Then, we selected the two fastest and two slowest movements for each subject (*Figure 3D*, left, individual points) and combined these movements across all subjects. Then we fit *Equation (2)* to both the slow movement and fast movement distributions. We tested for differences in integration gain by reporting the move duration by move force integral interaction effect on hold force within an ANCOVA.

## Reaching movements in a velocity-dependent force field

The serial architecture between moving and holding (*Figure 1A*) makes a strong prediction: external adaptation of moving activity will lead to changes in holding still. To test this idea, we used a force field paradigm. We gradually adapted reaching movements of a set of participants (n = 32) to a force field that exerted forces on the hand that were perpendicular and proportional to its velocity according to:

$$\begin{bmatrix} f_x \\ f_y \end{bmatrix} = b \begin{bmatrix} 0 & -1 \\ 1 & 0 \end{bmatrix} \begin{bmatrix} v_x \\ v_y \end{bmatrix} \tag{9}$$

Here, $v_x$ and $v_y$ represent the x and y velocity of the hand, $f_x$ and $f_y$ represent the x and y force applied to the hand by the robot, and $b$ represents the magnitude of the force field (in units of N·s/m). When $b > 0$, this corresponds to a clockwise (CW) field, and when $b < 0$, this corresponds to a counterclockwise (CCW) field.

In the experiment shown in *Figure 3*, participants were exposed to both CW and CCW force fields while making 10 cm movements. The experiment was structured so that the force field magnitude would start at 0 (a null field trial) and then gradually increase to its maximum strength over many trials. After this, the magnitude would then be reduced back to 0 over many trials. The exact perturbation schedule for healthy subjects is shown in *Figure 3D*. The experiment started with 40 trials (20 outwards movements and 20 backwards movements) of null field trials. Next, subjects were adapted to either a CW or CCW force field. The magnitude of CW/CCW perturbation was increased/decreased from 0 to 15 /- 15 N·s/m over the course of 100 outwards reaching trials (200 trials in total). The perturbation magnitude was then maintained at a constant level of 15 /- 15 N·s/m over the course of 50 outwards reaches (100 trials total) and then brought back to 0 gradually in a de-adaptation period of 100 outwards reaching trials (200 trials total). After this de-adaptation period, participants continued to reach in a washout period of 20 outwards reaches (40 trials total) where no force field was applied. Participants were then given a short break and this structure was repeated (either for the same force field, but a different target position (n = 17), or the opposite force field and the same target position (n = 15).

The experimental protocol was nearly identical (*Figure 3D*) for our stroke patient experiments, with two small differences. First, trial counts differed. Adaptation and de-adaption periods were 160 trials, as opposed to 200 trials. The period of maximal perturbation magnitude was reduced from 100 trials to 80 trials. The second difference is that we increased the maximal force field magnitude to 18 N·s/m. We increased this magnitude to compensate for slower movements (we required 600–

800 ms movement duration for stroke patients and their age-matched controls, but 450–550 ms in other experiments). Each arm was tested in the stroke experiments in four blocks: paretic, non-paretic, paretic, non-paretic. The perturbation magnitude went in an A-B-B-A order (where A and B refer to either a CW or CCW field). The arm was supported by a frictionless air sled for both patients and age-matched controls.

For both healthy subjects and stroke patients, we measured the relationship between the move period forces and the change in holding forces at regular intervals throughout the process of adaptation and de-adaptation. To do this, every 5th outwards reaching movement was performed in a channel. We measured the within-trial relationship between moving and holding forces (*Equation 2*).

We performed a control experiment (n = 7 subjects, *Figure 3—figure supplement 1*) to determine if holding forces were stable over longer periods of time. In this experiment, the length of the hold period was increased from approximately 1.8 to 6.5 s. To keep subjects engaged over this period of time, subjects were exposed to a working memory task during the holding period of channel trials as described in C3. The experiment was otherwise similar to the other tasks described in this section.

## Reaching movements in a zero integral force field

We considered an alternative hypothesis: holding forces could be a trivial continuation of moving forces (as opposed to an integral of moving forces). To test this idea, we designed two position-dependent force fields, A and B, with the latter integrating to 0 (*Figure 4*).

Subjects reached between two targets separated by 20 cm. To form a zero integral force field, we designed a perturbation with two components, $FF_1$ and $FF_2$. Each perturbation produced force along the x-axis, while movements were made along the y-axis. The first component $FF_1$ was applied during the first 10 cm of the reach, and the second component $FF_2$ was applied during the last 10 cm of the reach. Each component was programmed as a quadratic function of position. For $FF_1$ zero force was applied at the start position and at 10 cm. The maximal force was reached at 5 cm (the vertex of the first parabola). For $FF_2$ zero force was applied at 10 cm and at the target position. The maximal force was reached at 15 cm (the vertex of the second parabola). Here, we refer to the magnitude of $FF_1$ and $FF_2$ as the maximal force of each perturbation. To obtain a zero integral force field, $FF_1$ and $FF_2$ produced forces in opposite directions.

The experiment started with 25 outwards (50 trials total) null field trials ($FF_1$ and $FF_2$ were both equal to zero). Then we gradually increased the magnitude of $FF_2$ while $FF_1$ remained at zero (*Figure 4B*, Phase 1). $FF_2$ was increased from 0 N to 3.5 N in even increments, over the course of 100 outwards reaching movements (200 trials total). Then Phase 2 of the experiment started. In this phase, $FF_2$ was maintained at 3.5 N on all trials, while $FF_1$ was gradually changed. The magnitude of $FF_1$ was decreased from 0 N to $-3.5$ N over the course of 200 outwards reaching trials (400 trials total). In this way, at the end of the experiment, participants were exposed to two force fields within the same reaching movement that perturbed the hand in opposite directions but with equal magnitude. Throughout this paradigm, we measured moving and holding forces on every 5th outwards reach in a channel.

We found that holding forces gradually decreased in Phase 2 of the experiment, consistent with our hypothesis of integration. To make sure that the introduction of $FF_1$ caused this decrease in holding force, as opposed to repetition or fatigue, we performed a control experiment (n = 11, *Figure 4A*). In the control experiment, Phase 1 was identical to the main experiment described above. However, during Phase 2, the magnitude of $FF_1$ was maintained at 0 and the magnitude of $FF_2$ was maintained at 3.5 N. All other details were identical. We measured moving and holding forces (*Figure 4A*) on every 5th outwards movement in a channel.

## Measurement of the postural field

In order to hold the limb still, the postural controller generates a converging field of position-dependent forces that counters unwanted displacement of the limb (*Mussa-Ivaldi et al., 1985*; *Shadmehr et al., 1993*). In some of our experiments (*Figure 5*), we set out to measure this field. To do this, we designed a postural probe (*Figure 5A*). On a postural probe trial, the robot moved the hand slowly in a random direction after the hand stopped within the target. As the hand was moved,

visual feedback of hand position was prevented: the display cursor was frozen at the holding location.

To quantify the output of the holding controller, we measured the forces the subject applied to the handle while the robot moved the hand. To prevent participants from voluntarily opposing the imposed hand displacement, we distracted each participant with the working memory task described in an earlier section. We did not inform participants as to the nature or presence of the postural perturbation. Instead, we instructed participants to solely concentrate on the working memory task and obtain as many points as possible by answering memory questions correctly. Points for correct responses were combined with the points awarded for successful reaching movements.

The postural probe consisted of a straight-line displacement designed to make the probe as imperceptible as possible. To move the hand, we placed the hand in a two-dimensional clamp with stiff spring-like mechanics (stiffness = 4000 N/m, viscosity = 75 N·s/m) and moved the equilibrium position of the clamp through the workspace a total of either 2.5 cm, 4 cm or 5 cm, depending on the trial. The imposed motion consisted of three phases.

In Phase 1, the hand was moved a short distance (0.15 cm for 2.5 cm probes, 0.15 cm for 4 cm probes, and 0.3 cm for 5 cm probe) along a minimum jerk trajectory, over a short duration (0.75 s for 2.5 cm probes, 0.75 s for 4 cm probes, and 1.5 s for 5 cm probes). At the end of this displacement the velocity of the hand was equal to 0.375 cm/s. In Phase 2, the hand was then moved at this constant velocity for a specified displacement (2.2 cm for 2.5 cm probes, 3.7 cm for 4 cm probes, and 4.4 cm for 5 cm probes). This constant velocity displacement lasted for 5.87 s for 2.5 cm probes, 9.87 s for 4 cm probes, and 11.73 s for 5 cm probes. In Phase 3, the hand was slowed to rest over a short distance (0.15 cm for 2.5 cm probes, 0.15 cm for 4 cm probes, and 0.3 cm for 5 cm probe) along a minimum jerk trajectory, over a short duration (0.75 s for 2.5 cm probes, 0.75 s for 4 cm probes, and 1.5 s for 5 cm probes). Finally, an additional buffer period of 0.3 s was added after reaching the final displaced position, prior to the end of the probe trial. The total duration of the probe was therefore 7.67 s for 2.5 cm probes, 11.67 s for 4 cm probes, and 15.03 s for 5 cm probes.

Critically, as stated earlier, the participant was not provided position feedback during the postural probe. Instead, cursor feedback of hand position was frozen at the target. Therefore, at the end of the postural probe, there was a discrepancy between the location of the hand and the location of cursor feedback. To seamlessly reunite the hand with its cursor feedback without drawing the attention of the participant, we manipulated visual feedback during the following reaching trial; as the next reach was executed, we projected the cursor position onto the line connecting the frozen cursor position and the position of the next target. In this way, it appeared to the participant as if they were reaching perfectly straight between the start and target position. At the same time, we confined the motion of the hand to a straight line connecting its displaced position with that of the next target. When the hand entered the target, a small and brief force pulse was applied to move the hand to the center of the target at which point $x$ and $y$ feedback was reunited with the true hand position.

## Quantifying the null point and shape of the postural field

As the hand of the participant was moved by the robot during postural probe trials, the displacement of the hand was opposed by postural restoring forces (*Mussa-Ivaldi et al., 1985*; *Shadmehr et al., 1993*; *Figure 5B*). To mathematically characterize the two-dimensional field of restoring forces (i.e. the postural field), we fit a simple mathematical model (*Mussa-Ivaldi et al., 1985*) that treated the arm as a linear two-dimensional spring with a single equilibrium point:

$$\begin{bmatrix} F_x \\ F_y \end{bmatrix} = K \begin{bmatrix} x - x_{null} \\ y - y_{null} \end{bmatrix} \tag{10}$$

where $F_x$ and $F_y$ are the forces applied to the handle due to displacement of the hand from the null point of the system $(x_{null}, y_{null})$ to some position $(x, y)$. The stiffness matrix $K$,

$$K = \begin{bmatrix} k_{xx} & k_{xy} \\ k_{yx} & k_{yy} \end{bmatrix} \tag{11}$$

describes the magnitude and orientation of the stiffness field. We constrained $K$ to be a symmetric matrix (i.e. $k_{xy} = k_{yx}$). We fit this linear spring model to the postural restoring forces by identifying

the parameter set (five free parameters, $x_{null}$, $y_{null}$, $k_{xx}$, $k_{xy}$, $k_{yy}$) that minimized the sum of squared error between the hand forces (collapsed across the x and y axes) predicted by *Equation (10)* and the hand forces measured during all of the postural probe trials. For this fit, we used the forces measured within the ellipse bounded by −2.25 to 2.25 cm along the x-axis and −1.5 to 1.5 cm along the y-axis, relative to the final hand position. To locate the optimal parameter set, we used the genetic algorithm in MATLAB R2018a. We repeated the genetic algorithm search eight times and selected the one that minimized the squared error cost function. The optimal parameter set provided a good fit to the data, accounting for approximately 70% of the variance in the observed postural field ($R^2$ during baseline period, mean ± SEM: 0.70 ± 0.03; $R^2$ after adaptation: 0.69 ± 0.02).

To summarize the shape of the postural field, we considered three properties: (1) its null point, (2) its orientation, and (3) its stiffness. The null point was equivalent to $x_{null}$ and $y_{null}$. To calculate the orientation of the field, we considered the eigenvector of the stiffness matrix $K$ corresponding to the largest eigenvalue of $K$. We calculated the angle of this eigenvector in the x-y plane. To compute the stiffness of the field, we calculated the Frobenius norm of the stiffness matrix $K$.

## Measuring how changes in movement forces alter the postural field

To determine if changes in reaching forces altered the postural field, we measured the postural field before and after adaptation to a velocity-dependent force field. Subjects (n = 27) were adapted to a CCW velocity-dependent force field. To measure the postural field, postural probes moved the hand in 1 of 12 directions (0°, 15°, 30°, 45°, 90°, 135°, 180°, 225°, 270°, 315°, 330°, and 345° with respect to the x-axis) while participants were distracted with a working memory task.

We measured the postural field before and after adaptation. Before adaptation, participants completed 3 blocks of trials, each separated by a short break. In each block, all 12 postural probe directions were visited a single time. The probe displacement was 2.5 cm for all probe directions. Postural probes were given on every 4th outward reaching movement. Therefore, participants completed a total of 288 baseline trials (3 blocks x 12 probes/block x 4 reaching movement pairs/probe x 2 movements/reaching movement pair). Outwards reaching movements of 10 cm were all performed directly away from the body.

Participants were then gradually adapted to a velocity-dependent force field. The field magnitude was decreased from 0 to −10 N·s/m in constant increments over the course of 65 outwards reaching trials (130 trials total). After this adaptation period, the postural field was re-measured (*Figure 5B*, right). As before, all 12 probe directions were probed in a random order, three times. No breaks were provided in between blocks. We anticipated that the postural field would shift after adaptation to the force field. Therefore, we extended the probe displacement to 4 cm for probe angles of 0°, 15°, 30°, 45°, 315°, 330°, and 345°. Postural probes occurred at the same frequency as before adaptation for a total of 288 trials. To maintain participants in an adapted state, on all outwards reaching trials other than postural probe trials, a velocity-dependent perturbation was maintained at −10 N·s/m. Target-hold wells were applied to the final hand position during all outwards trials (with the exception of postural probe trials) during the holding period (1.5 s duration, with an addition 0.3–0.4 inter-trial-interval).

Our analysis focused on changes in the postural field due to adaptation. First, we looked for within-subject changes to the location of the null point of the field. Second, we looked for within-subject changes to the orientation and stiffness magnitude of the field. For visualization purposes, we constructed a two-dimensional postural field from the forces measured during probe trials using linear interpolation. To do this, along each probe direction we resampled forces in x and y spatially in intervals of approximately 0.1 cm. For each of the resampled restoring forces, we calculated the corresponding polar coordinates (i.e. the radius and angle). In polar coordinates, all x and y forces lied along a rectangular grid. We used bilinear interpolation along these polar coordinates to estimate the postural field within the space between the 12 probe angles.

## Measuring the relationship between holding forces and the null point of the postural field

We reasoned that the holding forces measured in our other experiments (*Figures 3*, *4*, *6* and *7*) were potentially caused by a misalignment between the hand (fixed in the channel at the target)

and the postural null point (somewhere displaced from the target). If this is true, we could gradually eliminate these forces if we displaced the hand toward its null point.

We recruited a set of subjects (n = 19) to test these predictions at several points during adaptation to a velocity-dependent force field. At regular intervals during adaptation, we inserted postural probe trials along 0°, with respect to the x-axis. This corresponded to the direction of the holding force. During the probe, participants were distracted with the working memory task. For the first 10 participants, we used 5 cm postural probe trials. For the last nine participants, we shortened the probe length to 4 cm. Here, we analyze only the first 4 cm of displacement to combine both versions of the experiment.

Before adaptation, we measured the postural forces a total of 10 times. Postural probes were inserted regularly on every 5th outward reach, within a baseline period of 100 trials (50 outwards and 50 backwards movements). Outwards reaching movements of 10 cm were all performed directly away from the body. Next, we adapted participants gradually to a CCW velocity-dependent force field, where we decreased the field magnitude from 0 to $-10.5$ N·s/m in constant increments over 175 outwards reaching trials (350 trials total). We measured moving forces, holding forces, and the response to the postural probe on every 5th outwards reaching movement.

We found that hand forces during the postural probe passed through a null point as the hand was displaced from its terminal position (*Figure 5E*). To determine the location of the null point of the arm on a trial-to-trial basis we fit a three-parameter exponential function to the hand forces as a function of hand displacement in the probe, and recorded its x-intercept. To do this, we first resampled subject forces spatially in increments of 0.05 cm. Next, to reduce noise inherent in the single-trial force measurements, we used a bootstrapping approach. On each trial, we randomly sampled subjects with replacement, calculated the mean postural force as a function of distance across the sample, and fit the exponential to this mean behavior. We repeated this process 2000 times, and used this distribution to estimate 95% confidence intervals around the mean (*Figure 5F*).

## Measuring adaptation in the gain of integration

To maintain stability at the endpoint, the integrator must adapt as the reach controller adapts. For the oculomotor system, adaptation of the integrator occurs when there are consistent errors between the terminal position of the eye, and its target (*Optican and Miles, 1985*). We speculated that such a mechanism might also adapt the gain of the reach integrator. To detect these errors, we considered the reach trajectories of participants as they adapted to velocity-dependent force fields (*Figure 6B*). We spatially aligned reach trajectories by subtracting off the terminal hand position. Next, we isolated the lateral component of the reach trajectory.

We observed that participants often exhibited deviations in their hand position as they attempted to stop their hand at the end of movement (*Figure 6B*, endpoint correction). To quantify the size of these errors, we measured the largest 'positive' deviation of the hand from its terminal position, after the hand exceeded 80% of its reach trajectory. On movements in which no such error occurred, we instead calculated the largest 'negative' deviation of the hand from its terminal position. In *Figure 6B*, we highlight these errors for two example participants by temporally aligning reach trajectories to the point at which the hand exhibited the largest 'positive' deviation from its terminal position.

In *Figure 6D*, we compare the magnitude and sign of the late trajectory deviations, to the gain of integration at the end of adaptation. To calculate the integration gain, we divided the median hold force by the median reach force integral, measured over 10 channel trials after the perturbation magnitude had plateaued (see horizontal line after CCW+ and CW+ in *Figure 3E*, bottom). Similarly, in *Figure 6E*, the integration gain was calculated by dividing the hold force by the reach force integral, either over the first one-third of CCW+/CW+ trials (for the FF early period) or the final 50% of CCW+/CW+ trials (for the FF late period). For the null period, we could not use this technique, as the mean reach force integral and the hold force were both nearly zero. Therefore, to calculate integration gain, we linearly regressed hold period force onto the integral of reach period force, and reported the slope of the regression line.

## Acknowledgements

This work was supported by grants from the National Institutes of Health (R01NS078311, 1DP2NS083037, R01NS100066, 1U19NS104649, F31NS095706, and F32NS092350), the National Science Foundation (1723967), the Simons Foundation (SCGB#542957), the UK Medical Research Council (MR/P023967 and MR/K023012/1), and the Sheikh Khalifa Stroke Institute. We thank Kahori Kita for help with performing and scoring the assessments for the stroke patients, and Jennifer Keller for help performing some of these assessments and allowing us to borrow dynamometry equipment. We also thank all study participants, especially stroke survivors, for their time and contribution to this work.

## Additional information

### Funding

| Funder | Grant reference number | Author |
| --- | --- | --- |
| National Institute of Neurological Disorders and Stroke | R01NS078311 | Reza Shadmehr |
| National Institute of Neurological Disorders and Stroke | 1DP2NS083037 | Mark M Churchland |
| National Institute of Neurological Disorders and Stroke | R01NS100066 | Mark M Churchland |
| National Institute of Neurological Disorders and Stroke | 1U19NS104649 | Mark M Churchland |
| National Institute of Neurological Disorders and Stroke | F31NS095706 | Scott T Albert |
| National Institute of Neurological Disorders and Stroke | F32NS092350 | Mark M Churchland |
| National Science Foundation | 1723967 | Reza Shadmehr |
| Simons Foundation | SCGB#542957 | Mark M Churchland |
| Medical Research Council | MR/P023967 | Stuart N Baker |
| Sheikh Khalifa Stroke Institute | | John W Krakauer |
| Medical Research Council | MR/K023012/1 | Demetris S Soteropoulos |

The funders had no role in study design, data collection and interpretation, or the decision to submit the work for publication.

### Author contributions

Scott T Albert, Conceptualization, Data curation, Software, Formal analysis, Supervision, Funding acquisition, Validation, Investigation, Visualization, Methodology, Project administration; Alkis M Hadjiosif, Data curation, Formal analysis, Funding acquisition, Validation, Investigation, Visualization, Methodology; Jihoon Jang, Data curation, Software, Formal analysis, Investigation, Visualization, Methodology; Andrew J Zimnik, Demetris S Soteropoulos, Resources, Data curation, Software, Investigation, Methodology; Stuart N Baker, Mark M Churchland, Resources, Data curation, Software, Supervision, Funding acquisition, Investigation, Project administration; John W Krakauer, Resources, Data curation, Supervision, Funding acquisition, Investigation, Project administration; Reza Shadmehr, Conceptualization, Formal analysis, Supervision, Funding acquisition, Investigation, Visualization, Methodology, Project administration

### Author ORCIDs

Scott T Albert  https://orcid.org/0000-0001-9140-1077
Alkis M Hadjiosif  https://orcid.org/0000-0001-8823-3631
Mark M Churchland  http://orcid.org/0000-0001-9123-6526

John W Krakauer [iD] http://orcid.org/0000-0002-4316-1846
Reza Shadmehr [iD] https://orcid.org/0000-0002-7686-2569

## Ethics

Human subjects: Informed consent was obtained from all participants. All human subjects work was approved by the Johns Hopkins School of Medicine Institutional Review Board, protocol number NA_00037510.

Animal experimentation: All animal procedures in the U.S. were conducted in accord with the US National Institutes of Health guidelines and were approved by the Columbia University Institutional Animal Care and Use Committee (AC-AAAQ7409). These data were originally published in Lara, Cunningham, & Churchland (2018) as well as Lara, Elsayed, Zimnik, Cunningham, & Churchland (2018). All procedures in the U.K. were carried out under appropriate UK Home Office licenses in accordance with the Animals (Scientific Procedures) Act 1986, and were approved by the Local Research Ethics Committee of Newcastle University. These data were originally published in Sotero-poulos, Williams, & Baker (2012).

## Decision letter and Author response

Decision letter https://doi.org/10.7554/eLife.52507.sa1
Author response https://doi.org/10.7554/eLife.52507.sa2

# Additional files

## Supplementary files

• Supplementary file 1. Measures of impairment in stroke patients. Patients completed the Fugl-Meyer Assessment (FMA) and the Action Research Arm Test (ARAT), as well as strength testing in the shoulder and elbow. Strength measurements were repeated twice and the maximal force was recorded on each effort and then averaged across repetitions. Two separate raters scored the FMA and ARAT assessments, and scores were averaged across raters. Missing entries in table indicate that the patient was unable to perform the desired action. Patients were selected based on MRI or CT scans, and/or available radiologic reports. Scans and/or reports were corroborated to determine the level at which the white matter of the corticospinal tract (CST) was lesioned. Here, we provide the level of the brain at which the white matter was lesioned.

• Transparent reporting form

## Data availability

Source data files generated or analyzed during this study are included for Figures 1-7 and have also been deposited in OSF under accession code YC64A.

The following dataset was generated:

| Author(s) | Year | Dataset title | Dataset URL | Database and Identifier |
| --- | --- | --- | --- | --- |
| Scott Albert | 2020 | Postural control of arm and fingers through integration of movement commands. | https://osf.io/yc64a/ | Open Science Framework, 10.17605/OSF.IO/YC64A |

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
