## [Decision Letter]

**Acceptance summary:**

The field of motor control typically assumes that the same cortical regions govern movement control and postural control of the arm. With a series of cleverly-designed behavioral experiments, the present study provides convincing evidence that the commands for holding the arm still depend on the integral of the motor commands that move the arm to the holding position. This new relationship has been shown in arm and finger movements, in monkeys and humans, and among stroke survivors. As the damage to descending cortical pathways spares this dependence between movement and posture, the postulated process of neutral integration might reside in subcortical areas. These new insights echo similar findings in eye movement control, suggesting a possible unifying principle for movement and posture control of different effectors. The study provides a new perspective for resolving the posture-movement paradox in the area of motor control. Furthermore, the findings will inspire researchers in motor learning (e.g., how can the integration function adapt to different situations?), neurophysiology (e.g., what are the neural underpinnings of proposed cortical and subcortical processing related to movement and posture, respectively?), brain-machine interface (e.g., what to decode from motor cortex at different phases of movement?), and medicine (e.g., a possible holistic view of movement and postural disorders among stroke survivors or individuals with dystonia).

**Decision letter after peer review:**

Thank you for submitting your article "Postural control of arm and fingers through integration of movement commands" for consideration by *eLife*. Your article has been reviewed by two peer reviewers, one of whom is a member of our Board of Reviewing Editors, and the evaluation has been overseen by Michael Frank as the Senior Editor. The reviewers have opted to remain anonymous.

The reviewers have discussed the reviews with one another and the Reviewing Editor has drafted this decision to help you prepare a revised submission.

Summary:

The relationship between postural control and movement control has puzzled movement scientists for long. The current study proposes a novel model for limb motor control with a surprising linear correlation of motor commands during movement and the postural holding period. The study includes a broad set of experiments using different animal and human paradigms. When monkeys were moving their limb against gravity and a spring load, the authors found a linear correlation between EMG integral during movement and a change of average EMG amplitudes during the holding period. Consistent with the integration model, the authors found that the hand force measured by the force channel during reaching and during the hold period around the target conformed to the same correlation. Stroke survivors who had damage to the corticospinal tract showed a similar integration function, hinting that the integration of movement commands may happen in some unknown subcortical system.

Both reviewers find the study provides a broad range of evidence supporting the proposed integration model. The paper is well written, and the analyses are appropriate overall with a few exceptions. The major criticism includes 1) the data could be explained by alternative explanations that are less attractive, and 2) the utility of the model might be limited, but it has not been fully explained.

Essential revisions:

1) The data from the two monkey experiments might have alternative explanations. The correlation between EMG integral and the difference in EMG between two hold periods might reflect biomechanical demands for moving and maintaining posture in a force field, as both reviewers pointed out. For moving against gravity in the vertical plane and moving against a spring force, the EMG activity for holding at the end posture is biomechanically determined by the end position, and the end position is determined by the muscle work (best characterized by EMG integral) during the movement. Thus, the integration function is expected, given only postural muscles are included for analysis. Similarly, reviewer 2 also pointed out that for moving against a spring load, different EMG magnitude will cause different displacement and lead to altered static force.

2) However, the EMG integral will only predict the EMG at the hold position if movement time is maintained constant. Reviewer 2 suggests to provide more data on this issue to address related questions, i.e., was movement time roughly constant? Was the integral relation less accurate for outlier trials?

3) Did movement-period EMG predict hold position and its EMG at the same time? If hold EMG was not correlated with position (force), what decoupled them? Variation in co-contraction? Variation in activity levels across redundant muscles?

4) Similarly, for ruling out the biomechanical explanation, reviewer 1 suggests performing a small experiment without force field, e.g., reaching towards horizontal targets with frictionless weight support. If the same integration function is found, the integration model will be more supported as a principal explanation for the relationship between movement control and postural control. As the paper is already lengthy with a variety of experiments, the authors might consider whether this suggested experiment can be skipped. If so, a full discussion about constraints and limitations of the current model, along with the additional analyses above, should be included in the revision.

5) The rationale of the force adaptation experiment is not clear.

6) Reviewer 2 realized that the object of having the partial return channels, the target-hold well, and gradual adaptation was to make the perturbation unnoticed to the subject. However, the purposes of these methodological designs were buried in Materials and methods. We thus suggest to explain it in the Results briefly.

7) Reviewer 1 pointed out that force integral or average force is equated to motor command by the authors without justification. Furthermore, the force measured by the force channel is perpendicular to the movement direction, while the propelling force for reaching is in the movement direction. Thus, the integration function only accounts for perpendicular deviations from the movement direction.

8) Currently, the integration model poses to be applicable to all movements. In a similar vein, reviewer 1 questioned whether the correlation between force integral and the holding force could be explained by a common positional deviation during and after the reach. We suggest explaining the rationale of data analysis and experimental design of the force field adaptation experiments, and/or making it clear at the outset that the model is only for countering positional deviations in a force field.

9) Related to the point 1 above, a discussion on how the proposed integration system of posture control could work in the real world is needed. Reviewer 2 pointed out that mass or moment of inertia changes on a moment-to-moment basis with limb posture during movement, and the impedance of the limb is also modulated when we interact with different objects. The changes in viscous or inertial loading affect the movement forces without affecting static/postural forces. These real-life situations are common, but they depart from the current contrived lab conditions and appear unpredictable by the current model. The authors must provide some plausible explanations for the utility of the current model.

---

## [Author Response]

Essential revisions:1) The data from the two monkey experiments might have alternative explanations. The correlation between EMG integral and the difference in EMG between two hold periods might reflect biomechanical demands for moving and maintaining posture in a force field, as both reviewers pointed out. For moving against gravity in the vertical plane and moving against a spring force, the EMG activity for holding at the end posture is biomechanically determined by the end position, and the end position is determined by the muscle work (best characterized by EMG integral) during the movement. Thus, the integration function is expected, given only postural muscles are included for analysis. Similarly, reviewer 2 also pointed out that for moving against a spring load, different EMG magnitude will cause different displacement and lead to altered static force.

As noted by the reviewers, the work done to move an object against a force, is related to the magnitude of its displacement. For gravity, the work should be proportional to displacement, and for a spring, the work should be proportional to the square of displacement. If work done by the muscles can be roughly approximated by the integral of muscle activity, then the correlations between moving and holding activity we observe might be a consequence of these biomechanical constraints.

In the revised manuscript we perform control analyses for both our reaching data (Figure 1) and finger movement data (Figure 2) to rule out each of these possibilities. First, we consider reaching movements. Here, we made use of the fact that the monkey performed movements to eight different directions. Two of these directions were entirely horizontal movements, and the other six directions contained a vertical component. If the relationship between hold activity and the integral of move activity was solely due to moving in the gravitational field, we would expect that the integration function would differ for these two classes of movements. However, this is not what we observed. The gain of integration was no different for horizontal movements in which little to no work is done against gravity, and vertical movements where much of the work is done against gravity (ANCOVA, movement type by moving EMG integral interaction effect on holding activity, F=0.37, p=0.54).

Therefore, it appears unlikely that the integration function we observed is solely caused by moving under the influence of gravity. We now have added this analysis to Figure 1.

We also considered reaching movements in humans with and without support of the arm against gravity. We discuss this comparison in point 4 below. Briefly, we found that integration was present with a gain that was indistinguishable for movements with and without weight support of the arm.

Next, we considered how biomechanical constraints may have impacted our analysis of finger movements in Figure 2. In the original manuscript, we noted that muscle activity during the hold periods was highly variable across trials, despite little trial-to-trial variability in the position of the finger. Nevertheless, as noted by the reviewers, it remains possible that trial-to-trial changes in the magnitude of the primary finger movement could have led to the trial-by-trial correlations in move and hold activity; movements with a larger displacement require more work (i.e., moving muscle activity) and terminate at a position with a larger spring force (necessitating more holding activity).

This alternate hypothesis, hinges on two relationships. One, trial-by-trial changes in hold position must correlate with trial-by-trial changes in hold EMG activity. Two, trial-by-trial changes in the finger displacement (i.e., movement size) must correlate with the integral of moving EMG activity. However, across the recorded muscles, both of these correlations were exceedingly weak. Variation in the final finger position accounted for less than 2% of the trial-to-trial variability in the final hold EMG activity (R^2^ = 0.0107 ± 0.0041, mean ± SEM, across all muscles in both monkeys, individual regressions for each muscle). We now address this point in the Results section. And secondly, variations in finger displacement accounted for less than 1% of the trial-to-trial variability in the integral of moving EMG activity (R^2^ = 0.006 ± 0.0018, mean ± SEM, across all muscles in both monkeys, individual regressions for each muscle). We now address this point in the Results section.

For comparison, we re-evaluated the integration hypothesis. Here, we found that approximately 40% of the variance in the change in hold EMG activity was predicted by trial-by-trial changes in the integral of moving EMG activity (R^2^ = 0.42 ± 0.031, mean ± SEM, across all muscles in both monkeys, individual regressions for each muscle).

Therefore, for both reaching movements and finger movements, differences in the physical constraints imposed by either gravity, or spring forces, did not appear to be viable alternative solutions to the integration hypothesis. We now discuss all of these control analyses in the figures and text of the revised manuscript.

2) However, the EMG integral will only predict the EMG at the hold position if movement time is maintained constant. Reviewer 2 suggests to provide more data on this issue to address related questions, i.e., was movement time roughly constant? Was the integral relation less accurate for outlier trials?

As noted, time is another dimension along which integration can be tested. Unfortunately, for the finger movement data, movement time was carefully constrained in this tracking task. Trials with errors larger than 1.4° at any point in the movement were immediately aborted. This served to strongly constrain movement time and movement kinematics, so there are no outlier trials which can be analyzed to address the reviewer’s suggestion.

However, in our reaching experiment, movement time was less constrained. Although the monkeys tended to reach at a stereotypical speed, this speed fluctuated across sessions. Therefore, to test how time affected integration, for each movement direction, we split our data about the median movement speed. We collected all muscles recorded on slower movement sessions, and all muscles recorded on faster movement sessions. We performed separate regressions for each of these groups, and for each movement direction. The same integration gain was observed for both types of movements (two-sample t-test, p=0.30).

These results should be taken with some caution however. The duration of fast movements was roughly 350 ms (350.3 ± 11.1 ms, mean ± SEM) and the duration of slow movements roughly 450 ms (453.6 ± 4.9 ms, mean ± SEM); slower movements were only 30% slower than fast movements. Therefore, we cannot be sure that the integration gains reported here would extend to vastly different movement durations. But it does appear that the same integration gain applies across roughly 30% differences in movement duration. We now report this control analysis in Figure 1.

We also performed a similar analysis for our human reaching data. For each subject in the null period, we sorted their channel trial movements according to movement duration. Then we selected the two fastest reaches and two slowest reaches for each participant, and collapsed these into a distribution of fast movements (Figure 3, red dots at left) and a distribution of slow movements (Figure 3, black dots at left).

For each of these distributions we linearly regressed the change in hold forces onto the integral of reach forces. At right we show the 95% confidence intervals for the regression slope, i.e., the integration gain. Even though fast and slow movements differed by approximately 30% in duration (see middle inset, fast movement duration of 0.49 ± 0.05 sec and slow movement duration of 0.66 ± 0.05 sec, mean ± SD), we found no difference in the gain of integration (ANCOVA, movement type by move force integral interaction effect on hold force, F=0.007, p=0.935). We now report this control analysis in Figure 3.

3) Did movement-period EMG predict hold position and its EMG at the same time? If hold EMG was not correlated with position (force), what decoupled them? Variation in co-contraction? Variation in activity levels across redundant muscles?

As we noted in our response to point 1 above, we did not observe a correlation with either movement-period EMG or hold period EMG, and the associated hold displacements or positions. Generally, differences in hold position accounted for less than 2% of the variability in either hold period or move period EMG.

As noted by the reviewer, one possibility that could explain both the correlations between move and hold EMG, and the lack of dependence on hold position, is co-contraction. If on some trials, the finger was stiff and co-contracted its muscles, and on other trials less so, we may observe correlations between move period and hold period activity, without any relation to hold position. To change finger stiffness, agonist and antagonist pairs of muscles would exhibit correlated increases or decreases in activity. In other words, according to this hypothesis, we should be able to predict the hold period activity in one muscle not solely based on its move activity (as in the integration hypothesis) but also on the activity of simultaneously recorded muscles. To test this idea, we regressed hold period activity in each muscle onto the integral of move period EMG in other muscles. We found that roughly 10% of the variability in hold EMG could be explained by the integral of move activity of other muscles (R^2^ = 0.10 ± 0.02, mean ± SEM, across all muscles in both monkeys, individual regressions for each muscle-muscle pair).

For comparison, recall that the integral of move EMG predicted approximately 40% of the variance of hold activity in the same muscle. Therefore, while some of the variance in hold activity was indeed shared across the move activity of other muscles (consistent with co-contraction), move activity in a given muscle remained a much better predictor of hold activity in that same muscle (consistent with integration). We now note these points in the Results.

It is important to note that while finger movements were constrained to one degree of freedom using a narrow tube, it is likely that the true equilibrium position of the finger on any given trial, did not lie exactly along the constrained movement axis, but somewhere in three-dimensional space. In other words, like our reaching experiments in Figure 5, if the narrow tube were removed, the finger would likely have moved to a slightly different terminal position. This might explain why muscle activity only poorly correlated with the final position of the finger, even under the spring load. Therefore, under the integration hypothesis, we speculate that variability in activity during the finger movement would integrate to different levels of hold activity, thus leading to slight deviations in the final hold position. Such an interpretation is consistent with the trial-by-trial integration of natural movement variability we observe for reaching movements in Figure 3B and C. We describe this possible interpretation in the Discussion.

Taken together, our combined results from points 1, 2, and 3, suggest that while co-contraction and movement against spring-like forces can explain part of the observed relationship between moving and holding activity, the integration hypotheses remains the best descriptor of our primary findings.

4) Similarly, for ruling out the biomechanical explanation, reviewer 1 suggests performing a small experiment without force field, e.g., reaching towards horizontal targets with frictionless weight support. If the same integration function is found, the integration model will be more supported as a principal explanation for the relationship between movement control and postural control. As the paper is already lengthy with a variety of experiments, the authors might consider whether this suggested experiment can be skipped. If so, a full discussion about constraints and limitations of the current model, along with the additional analyses above, should be included in the revision.

We thank the reviewers for this suggestion. We collected reaching data with and without arm support. We placed the arm on a frictionless air sled to support its weight for both our stroke patients and associated control participants (Figure 7). Consistent with our integration hypothesis, we found that the integration gain did not differ across the supported and unsupported conditions (p=0.90, see Figure 3).

Therefore, it does not appear that the correlations between moving and holding activity are a trivial consequence of biomechanical demands imposed by gravity. This analysis further supports our response to point 1 above, where we show no difference in the integration for vertical and horizontal reaching movements of the arm. We now address this important control in Figure 3.

5) The rationale of the force adaptation experiment is not clear.

Thank you for this important feedback. The integration hypothesis makes a more important prediction: if there is a change to the move commands, there should be a change in hold commands, even if the holding location remains the same. We used force fields as a means of testing this idea. By perturbing reaching movements with force fields our goal was to bias the moving commands in different directions through the process of adaptation. If holding commands are computed from the moving commands, biases in moving forces should lead to changes in holding commands even though (1) we never perturbed the holding commands and (2) the holding position was invariant throughout the entire process of adaptation. In this way, we feel that our data in Figures 3-7 support a more causal role of movement integration in the control of holding still.

We appreciate that this rationale was not well-motivated in the text. We have revised the subsection “Hold period activity for finger movements” to better explain our thought process.

6) Reviewer 2 realized that the object of having the partial return channels, the target-hold well, and gradual adaptation was to make the perturbation unnoticed to the subject. However, the purposes of these methodological designs were buried in Materials and methods. We thus suggest to explain it in the Results briefly.

We thank the reviewer for their careful review of our methods. We agree that we could make these more apparent in the main text. The reason we made these methodological choices was not to mask adaptation per se (though it does help in that regard), but instead to prevent the process of reach adaptation from impacting periods of holding still. Because our goal was to adapt moving commands, without perturbing holding commands, we attempted to prevent our perturbations and assays from impacting the period of holding still. The three constructs identified by the reviewer were intended to do just that.

1) We used the target-hold well (we now use this terminology in place of endpoint clamp to distinguish it from channel trials) to prevent subject holding forces from moving the hand off the target, thereby resulting in errors in the process of holding still.

2) At the end of the hold period, participants actively reached back to the start position. We realized that suddenly removing the target-hold well prior to this return reach could also cause a sudden displacement of the hand (for the same reason as point 1 above). Therefore, we used a partial channel trial for the return movement, so that the hand would continue to be supported in the perpendicular direction as the subject transitioned from holding at the target to reaching back to the start.

3) Finally, we used gradual adaptation to keep errors small throughout training. Large feedback responses to within-movement errors could lead to more errors in the stabilization of the hand within the target at the endpoint.

We now describe these points more clearly in the rationale for our force field experiments in the Results section.

7) Reviewer 1 pointed out that force integral or average force is equated to motor command by the authors without justification. Furthermore, the force measured by the force channel is perpendicular to the movement direction, while the propelling force for reaching is in the movement direction. Thus, the integration function only accounts for perpendicular deviations from the movement direction.

The reviewer’s criticism is well-taken. Unlike our non-human primate experiments, we did not record EMG during our human experiments. Instead, we measured changes in the forces that participants exerted against the handle. These forces served as a low-dimensional abstraction of the motor commands sent to the arm muscles. Therefore, the integral of reach force serves as a rough kinetic analogue of the integral of moving EMG activity (Figures 1 and 2). The fact that EMG signals and forces vary roughly linearly with one another, might explain why the gain for EMG integration was similar to that of force integration.

Critically, we designed our perturbations so that the learning dimension was orthogonal to the primary movement direction – the robot forces were perpendicular to the trajectory of the hand. This way, we could cleanly isolate the component of the motor commands that varied in response to the perturbation, from the motor commands responsible for the primary movement.

Because adaptation occurred along the perpendicular axis, this dimension served as the principal dimension along which to evaluate the integration hypothesis. In other words, because participants modified their perpendicular forces, but not their parallel forces, we would expect only changes to the holding commands along the perpendicular direction. In support of this idea, when we measured how reach adaptation altered the postural field in Figure 5, we observed that the null point of the postural field shifted along the perpendicular axis exclusively, consistent with our integration hypothesis.

Another benefit of using the force integral measure, was that it provided a signed measure to test if motor commands that pushed the arm left (i.e., reaching forces to the left) canceled out the motor commands that pushed the arm right (i.e., reaching forces to the right), in a manner consistent with integration. We tested this idea with our experiment in Figure 4.

We now more clearly state that our force measurements describe a component of the total motor command (the component along the adaptation direction). While we do not feel that our integration hypothesis solely accounts for this perpendicular dimension (i.e., we observe integration in the entire EMG signal for reaching and finger movements in Figures 1 and 2), we now note in our Discussion that measurement of EMG during the process of reach adaptation would provide a means of further testing the integration model.

8) Currently, the integration model poses to be applicable to all movements. In a similar vein, reviewer 1 questioned whether the correlation between force integral and the holding force could be explained by a common positional deviation during and after the reach. We suggest explaining the rationale of data analysis and experimental design of the force field adaptation experiments, and/or making it clear at the outset that the model is only for countering positional deviations in a force field.

We thank the reviewer for noting this concern. Recall that moving and holding forces are only measured on channel trials, where the motion of the hand deviates by less than 1 mm from a straight path connecting the start and target position (see example trajectories in Figure 3A). Therefore, perpendicular deviations on channel trials are rather small (i.e., <1% of the total hand displacement from start to target). Furthermore, these deviations are not the result of the force field, but rather the subject’s own forces. Therefore, if holding forces arise to counteract positional deviations, it is entirely counterproductive to produce them on the channel trials (where we measure such forces).

With that said, in line with the reviewer’s comment, it is important to consider the possibility that holding forces arise from a process of adaptation driven by perpendicular deviations that occur on the intervening force field trials. In other words, perhaps the small positional deviations on the gradual force field trials drive a process of adaptation in both moving and holding forces, thus leading to the observed correlations.

Fortunately, we can be confident that this possibility does not explain our results. Recall that in addition to our force field experiments, we also describe the relationship between moving and holding forces on channel trials interspersed within null field periods where there were never any perturbations (Figure 3B, C, Figure 7C, F left column, and H). We specifically measured forces during these periods to establish that the correlations between moving and holding forces are present for normal reaching movements that were never perturbed by a force field. Therefore, common positional deviations related to the force field did not cause the observed correlations between moving and holding forces. And by extension, it is not true that our model applies only for countering positional deviations in a force field. We re-emphasize this point in the Results section.

With that said, it does appear that positional deviations do lead to adaptation of postural control. However, this adaptation process acts to eliminate the holding forces, rather than increase them. We describe this in much more detail in point 9 below and in Figure 6 of the revised manuscript. In brief, positional errors that occur near the endpoint of the reaching movement require a correction that is oriented opposite to the observed holding forces. For this reason, the occurrence of these errors teaches the integrator to reduce its gain, thus leading to smaller holding forces. Please see point 9 below for a more comprehensive discussion.

Overall, we hope that our revisions in the subsection “Hold period activity for finger movements” better explain the rationale of data analysis and experimental design of the force field adaptation experiments, as suggested. And in addition, we hope our addition of Figure 6 better address the relationship between positional errors and the process of neural integration.

9) Related to the point 1 above, a discussion on how the proposed integration system of posture control could work in the real world is needed. Reviewer 2 pointed out that mass or moment of inertia changes on a moment-to-moment basis with limb posture during movement, and the impedance of the limb is also modulated when we interact with different objects. The changes in viscous or inertial loading affect the movement forces without affecting static/postural forces. These real-life situations are common, but they depart from the current contrived lab conditions and appear unpredictable by the current model. The authors must provide some plausible explanations for the utility of the current model.

We have now significantly revised our Discussion section to better address the role of the integrator in the larger context of postural control. We feel that our data support the idea that movements are encoded by the cortex and then integrated over time by an unknown subcortical area. This subcortical area then generates the commands required to stabilize the arm, holding it still after movement ceases.

One of the major advantages of this control scheme is that this hypothetical neural integration pathway would provide a feedforward mechanism by which reach commands could be used to predict the hold commands required to stabilize the terminal position of the arm, overcoming the instabilities that would arise from delays in sensory feedback.

There are a number of predictions that must be true if such a feedforward integration system were to maintain endpoint stability in real world scenarios. As noted by the reviewers, motor commands required to move the arm to a target position are not fixed, but vary across a range of conditions: (1) interactions between the arm and external objects, (2) interaction torques that arise when the body is in motion, and (3) even over time as our bodies grow. We know that the move system continuously adapts to these novel dynamics. A similar process of adaptation would also be required of the neural integrator.

In fact, this point is well-illustrated by our data. Consider the puzzling scenario presented in Figure 5 of the manuscript. In the presence of a force field, the reach controller readily adapts and changes the reach forces in order to steer the hand to the target. However, downstream of the reach controller, changes to the reach forces are integrated and cause the hold system to program an entirely different null position, creating a discrepancy. This implies that postural stability will be compromised in the face of an adapting reach controller.

To solve this problem, the integrator must also be adaptive: the integration function must change when there is an error between its current output and the desired movement endpoint (Figure 6A). In the oculomotor system, adaptation of the integrator is driven by consistent errors between the terminal eye position and the location of the target (Optican and Miles, 1985). In the revised manuscript, we report a similar mechanism by which the reach integrator also appears to adapt. We address this in a new figure added to the manuscript, Figure 6.

In Figure 6, we demonstrate that reach trajectories during adaptation to the gradual force field exhibit two different types of errors (Figure 6B). The first error comes midway through the movement. This “negative” error occurs due to incomplete compensation for the force field; this is the error that drives the process of reach adaptation. The second error comes late in the movement, just before the hand stops within the target. This “positive” deviation (labeled endpoint correction in Figure 6B) occurs in the direction opposite the shift in null point in Figure 5. Therefore, this deviation could serve as an error signal for adaptation of the neural integrator.

In support of this idea, we observed that individuals with larger perpendicular errors near the end of the movement (x-axis in D) also exhibited a smaller integration gain (y-axis in D; also see the absent hold forces in C for the subject with larger endpoint errors in B). Here, the errors were measured during the process of adaptation. The gain was measured at the end of adaptation. Therefore, about 40% of the variance in the final integration gain was explained by differences in perpendicular errors near the end of movement. If these changes in integration gain were driven by a gradual learning process, we would expect that the integration gain should change during the adaptation period, as errors are experienced repeatedly. To test this idea, we measured the integration gain at several points during the experiment. Indeed, as time progressed, the gain of integration decreased (repeated measures ANOVA, F=12.60, p<0.001). Critically, the integration gain was no different between the null field period and the initial part of adaptation (Figure 6E, post-hoc comparison, p=0.88). In other words, initially, the forces produced to counteract the force field were not integrated differently than the forces that arose from natural movement variability in the null field period. However, later in adaptation, the integration gain decreased (Figure 6E, post-hoc comparison, p<0.001), as late positional deviations were repeatedly encountered on force field trials.

Altogether, these data suggest that Equation 2 alone cannot predict the magnitude of holding forces. The gain of integration changes over time as the postural controller encounters errors between the desired holding location and the output of the putative integrator. This process of adaptation would be necessary to maintain integration accuracy with dynamical changes to the arm. We describe these new experiments and analysis in the Results and Materials and methods sections. In addition, we relate our proposed mechanism of adaptation to that observed for the adaptation of the oculomotor integrator in the Results.

In our Discussion, we emphasize how this adaptation mechanism is critical for the utility of the integrator in real world scenarios (Results). In addition, we re-emphasize that the control of posture is not a task that would be handled by the neural integrator alone. There are a number of layered controllers that together, would provide for more robust control of endpoint stability. These include visual corrective systems in the cortex, and proprioceptive controllers in the spinal cord and cortex. We describe these ideas in the Results and Discussion sections, and relate them to our knowledge about the control of head movements, which we know both uses a neural integrator in the interstitial nucleus of Cajal, and a separate proprioceptive controller.

We hope that our changes to the manuscript (Results) more clearly address the utility of the proposed model, and provide a better description for how the integration system may fit into the larger context of postural control.